# On the Power of Source Screening for Learning Shared Feature Extractors

Leo Muxing Wang [1]   Connor Mclaughlin [1]   Lili Su [1]

## Abstract

Learning with shared representation is widely recognized as an effective way to separate commonalities from heterogeneity across various sources. Most existing work includes all related data sources via simultaneously training a common feature extractor and source-specific heads. It is well understood that data sources with low relevance or poor quality may hinder representation learning. We seek a deeper understanding of whether the inclusion of certain otherwise beneficial sources may inadvertently impair representation quality. For tractability, we develop the main theory in linear models and shared linear subspace, and further show that the same source-screening principle extends to nonlinear models. We find that source screening can play a central role in statistically optimal subspace estimation. Specifically, we show that, for a broad class of problem instances, training on a carefully selected subset of sources suffices to achieve minimax optimality, even when a substantial portion of data is discarded. Towards this, we introduce the notion of an *informative subpopulation* and develop algorithms, along with practical heuristics, to identify such subsets. We validate the effectiveness of our approach through theoretical analysis and empirical evaluations on both synthetic and real-world datasets.

## 1. Introduction

Training models from scratch is often inefficient in both data and computation, as it requires repeated relearning of similar low- and mid-level features. Recent progress in machine learning has been driven by the emergence of powerful general-purpose feature extractors that capture latent commonalities while separating heterogeneity (Bengio et al.,

2013; LeCun et al., 2015; Caruana, 1997; Collins et al., 2021). Specifically, via exploiting underlying relatedness, multi-task learning trains multiple tasks jointly, promoting information sharing across tasks and knowledge transfer to new ones (Caruana, 1997; Ando et al., 2005). Foundation models, the powerhouse behind recent AI advances, are trained at scale on heterogeneous and multi-source data to encode general knowledge across domains (Bommasani, 2021). Similarly, in federated learning, a parameter server learns a global model or shared feature extractor that enables client-specific model personalization (Fallah et al., 2020; Collins et al., 2022; McMahan et al., 2017; Kairouz et al., 2021; Collins et al., 2021).

However, a rigorous understanding of how to obtain a general-purpose feature extractor remains underdeveloped, even in linear settings (Niu et al., 2024; Crawshaw, 2020). With heterogeneous data sources/clients, learning effectiveness is not guaranteed. Negative transfer has long been observed to be a challenging empirical phenomenon (Zhang et al., 2022; Yang et al., 2025) in traditional single-source or multi-source transfer learning settings, and was not well characterized until recently. In shared representation learning, negative transfer is more subtle. Heuristic intuition suggests that data sources with low relevance or poor quality may impede representation learning. However, it is unclear whether relevance and data quality alone fully capture the mechanisms that govern negative transfer in shared representations. A fundamental question naturally arises:

*Question: Can the inclusion of otherwise beneficial sources inadvertently impair shared representation learning, and if so, which sources should be retained?*

Addressing this question in full generality is challenging given its breadth. To make progress, we first restrict our attention to linear models and study the problem of shared linear subspace learning, and then extend the obtained results to nonlinear models.

To obtain insights beyond trivial or degenerate cases, we further focus on *regression problems*. We consider a challenging regime in which all sources would traditionally be considered "good," in the sense that individual sources exhibit comparable relevance and quality with respect to the underlying common structure.

[1]Northeastern University. Correspondence to: Leo Muxing Wang <wang.muxin@northeastern.edu>.

*Proceedings of the 43rd International Conference on Machine Learning*, Seoul, South Korea. PMLR 306, 2026. Copyright 2026 by the author(s).

Implicitly assuming no "adversarial data sources", most existing work on shared representation training incorporates all available related data sources/clients via minimizing the (weighted) average loss:

$$\min_{\phi \in \Phi} \frac{1}{M} \sum_{i=1}^{M} \min_{h_i \in \mathcal{H}} F_i(h_i \circ \phi), \qquad (1)$$

where $M$ is the number of sources/clients, $F_i$ is the model prediction loss evaluated on source/client $i$, $\theta_i = (h_i \circ \phi)$ is the model that is decomposed into a shared feature representation $\phi$ and the source/client-specific head $h_i$ (Pan & Yang, 2009; Bengio et al., 2013; Finn et al., 2017; Thaker et al., 2023; Fallah et al., 2020; Collins et al., 2021), $\Phi$ is the representation class and $\mathcal{H}$ is the class of source-specific heads. This formulation effectively treats all incorporated sources as equally beneficial for representation learning, an assumption that often breaks down in practice.

Let $[M] = \{1, \cdots, M\}$ denote the collection of sources/clients, and $\mathcal{S} \subseteq [M]$ be an arbitrary subpopulation. Let $n_i$ denote the number of samples of source $i$. Motivated by the above gap, we study the following technical question:

*Technical question: Does there exist a subpopulation $\mathcal{S} \subseteq [M]$ such that learning a shared representation from $\sum_{i \in \mathcal{S}} n_i$ samples is not only more accurate than using all $\sum_{i=1}^{M} n_i$ samples, but also attains statistical optimality?*

**Contributions.** Our main contributions are summarized as follows. We primarily develop the theory under linear models with shared linear subspace, and further extend the resulting source-screening principles to nonlinear settings.

- We show that, for a class of problem instances, a state-of-the-art subspace estimator achieves minimax statistical optimality when trained on a suitably chosen subset of sources, even if a large fraction of sources is discarded.
- We formalize the notion of a desired subpopulation of sources and prove that training exclusively on such a subpopulation attains the minimax statistical optimality.
- We develop an efficient algorithm that provably identifies a good subpopulation in the genie-aided setting, and propose principled heuristics for subpopulation selection in the absence of genie information.
- Beyond the linear setting, we show that the source-screening principles extend to nonlinear models.
- We empirically validate the effectiveness of data source pre-screening and our proposed methods on both synthetic and real-world datasets.

## 2. Related Work

**Representation Learning.** Representation learning has been widely adopted for few-shot learning tasks. Most relevant recent works (Du et al., 2021; Collins et al., 2021; Thekumparampil et al., 2021; Duchi et al., 2022; Niu et al., 2024; Tian et al., 2025) focus on learning a low-dimensional shared representation across different tasks or data sources. (Du et al., 2021) considers $T$ source tasks and tries to learn a target task with much fewer data points. Their result shows full data utilization across all source tasks for the representation learning of the target task. (Tripuraneni et al., 2021) works under a similar problem setting but focuses on providing statistical rates for efficient algorithms. They also provide a minimax lower bound for recovering the subspace. (Thekumparampil et al., 2021) studies an alternating gradient-descent minimization method for subspace estimation, which achieves a near-optimal statistical rate. (Collins et al., 2021) proposes a federated learning framework and algorithm to learn the low-dimensional shared linear subspace, which is proven to converge to the ground truth representation with near-optimal sample complexity in a linear setting. (Duchi et al., 2022) provides a statistical estimator for the subspace and proves an upper bound for the estimation error under the setting that the noises are heterogeneous across clients. (Niu et al., 2024) provides a split-averaging estimator and establishes new upper and lower bounds for the estimation error of the low-dimensional subspace. They also extend the results to nonlinear models. (Zhang et al., 2024) proposes an adaptation of the alternating minimization-descent algorithm for non-i.i.d. and non-isotropic covariates, and establishes a linear convergence to the ground truth.

**Multi-task Learning.** In multi-task learning, task-relatedness is modeled by introducing structured coupling among task-specific predictors (Crawshaw, 2020). Early work assumes hard parameter sharing, where task predictors $\theta_i := (w_c, w_i)$ are decomposed into common parameters $w_c$ and task-specific parameters $w_i$, enforcing strong inductive bias (Caruana, 1997). Soft parameter sharing relaxes this assumption by considering $\theta_i$ in the form of $\theta_i = \widetilde{\theta}_0 + w_i$ and regularizing pairwise parameter differences $\|\theta_i - \theta_{i'}\|$ in training (Evgeniou & Pontil, 2004). (Lee et al., 2016) assumes one task's parameter can be succinctly expressed as a linear combination of other tasks' parameters. Their approach aims to select only the most relevant task relationships and suppress transfer between unrelated tasks by enforcing non-negativity and sparsity constraints on the regularization graph. Progressive Neural Networks (Rusu et al., 2016) considers primary tasks and auxiliary tasks. (Standley et al., 2020) proposes a framework that partitions tasks into clusters. Specifically, for a task set $\mathcal{T}$, there are $2^{|\mathcal{T}|} - 1$ combinations of partitions. Instead of exhaustive search, (Fifty et al., 2021) studies that when training all tasks in a single neural network, how does one task's gradient update affects other tasks' loss. By quantifying this inner-task affinity, they can find close to optimal auxiliary tasks. (Zhang et al., 2023) focuses on the task competition problem in

MTL, that is, certain tasks dominate the learning process and degrade the performance of other tasks. (Du et al., 2024) further investigates the parameter-level competition across tasks. They introduce PCB-MERGING (Parameter Competition Balancing) that drops parameters with low importance score, leading to improved performance across domains, number of tasks, model sizes, etc. Further details on MTL can be found in (Crawshaw, 2020).

**Client Selection in Federated Learning (FL).** Our work is also related to client selection in FL. A couple of key differences are: (1) we study training a shared representation rather than minimizing the average of local cost, and (2) we perform one-shot prescreening rather than active selection during training. Details are deferred to Appendix A.

# 3. Problem Setup

Up to this point, we have used the terms "source" and "client" somewhat interchangeably. Henceforth, for ease of exposition, we will use "source" only.

Similar to previous work (Tripuraneni et al., 2021; Collins et al., 2021; Thekumparampil et al., 2021; Duchi et al., 2022; Du et al., 2021; Tian et al., 2025; Niu et al., 2024), we consider a widely studied linear setup in which data from each source are generated by a linear model, and the models across sources share a low-dimensional linear subspace (i.e., the feature extractor $\phi$ in Eq. (1) is a low-dimensional matrix). We formally describe the setup next. Section 5 extends our results to nonlinear models.

We consider a learning system that consists of a parameter server and $M$ sources, where each source $i$ observes $n_i$ data points $\{(x_{ij}, y_{ij})\}_{j=1}^{n_i}$. Let $N := \sum_{i=1}^{M} n_i$ denote the total sample size. The parameter server cannot access the data. For source $i \in [M]$ and sample $j \in [n_i]$, $x_{ij} \in \mathbb{R}^d$ is the covariate vector, and $y_{ij} \in \mathbb{R}$ is the response generated by

$$y_{ij} = x_{ij}^\mathsf{T} \theta_i^\star + \xi_{ij}, \qquad (2)$$

where $\theta_i^\star \in \mathbb{R}^d$ is the ground-truth parameter for source $i$ and $\xi_{ij} \in \mathbb{R}$ is an additive noise. Let $\Gamma_i$ be the *unknown* covariance matrix shared by the covariates $\{x_{ij}\}_{j=1}^{n_i}$ at source $i$, satisfying $\mathbb{E}[x_{ij} x_{ij}^\mathsf{T}] = \Gamma_i$ for all $j$. In particular, the weighted parameters $\Gamma_i \theta_i^\star$ lie in a shared subspace of dimension $k \leq d$, spanned by the columns of an orthonormal matrix $B^\star \in \mathbb{R}^{d \times k}$. Then each source $i$ has its specific low-dimensional parameter $\alpha_i^\star \in \mathbb{R}^k$, such that

$$\Gamma_i \theta_i^\star = B^\star \alpha_i^\star. \qquad (3)$$

Here, $B^\star$ corresponds to $\phi$, and $\alpha_i^\star$ corresponds to source-specific head $h_i$ in Eq. (1). Henceforth, for ease of exposition, we refer $\alpha_i^\star$'s as *local heads*.

It is worth noting that $B^\star$ is not identifiable without sufficient diversity across $\{\alpha_i^\star\}_{i=1}^{M}$. If the local heads span

only a $k'$ dimensional subspace (where $k' < k$), then only partial $B^\star$ is revealed by these source tasks. Any missing directions cannot be recovered, and a downstream task lying in those directions would not benefit from the learned representation. This motivates us to investigate a key factor in determining the learnability of $B^\star$ in this problem (Du et al., 2021; Tripuraneni et al., 2020; 2021; Collins et al., 2021; Thekumparampil et al., 2021; Tian et al., 2025; Zhang et al., 2024; Niu et al., 2024), which is the spectrum of the matrix

$$D = \frac{1}{M} \sum_{i=1}^{M} \alpha_i^\star (\alpha_i^\star)^\mathsf{T}, \qquad (4)$$

This matrix $D$ captures the diversity of local heads $\alpha_i^\star$. Let $\lambda_r, r \in [k]$ denote the $r$-th largest eigenvalue of $D$. Under a set of standard assumptions, such as sub-Gaussian noises, sub-Gaussian covariates, and $\|\alpha_i^\star\| = O(1)$ for $i \in [M]$ (formally stated in Appendix B), prior studies (Tripuraneni et al., 2021; Du et al., 2021; Collins et al., 2021; Thekumparampil et al., 2021; Chua et al., 2021; Duchi et al., 2022; Duan & Wang, 2023; Tian et al., 2025; Zhang et al., 2024) have analyzed the statistical error rates for learning $B^\star$, measured by the principal angle distance.

**Definition 1** (Principal angle distance). Let $B, B^\star \in \mathcal{O}^{d \times k}$ be orthonormal matrices, where $\mathcal{O}^{d \times k} := \{B \in \mathbb{R}^{d \times k} : B^\top B = I_k\}$. Then the principal angle distance between $B$ and $B^\star$ is defined by:

$$\| \sin \Theta(B, B^\star) \| = \| B B^\mathsf{T} - B^\star (B^\star)^\mathsf{T} \|,$$

where $\| \cdot \|$ denotes the spectral norm.

For the general case, i.e., no additional conditions on $D$, the upper bound for the estimation error of $B^\star$ in (Tripuraneni et al., 2021; Du et al., 2021; Duchi et al., 2022; Duan & Wang, 2023) is of order $O\left(\sqrt{\frac{d}{N\lambda_k^2}}\right)$. More recently, (Niu et al., 2024) establishes a refined upper bound of $O\left(\sqrt{\frac{d\lambda_1}{N\lambda_k^2}} + \sqrt{\frac{Md}{N^2\lambda_k^2}}\right)$, which improves upon prior results in certain parameter regimes, such as in the well-represented setting where $\lambda_1 = \Theta(\lambda_k) = \Theta(1/k)$. Furthermore, (Tripuraneni et al., 2021) establishes a lower bound of order $\Omega\left(\sqrt{\frac{1}{(N\lambda_k)}} + \sqrt{\frac{dk}{N}}\right)$, which is is further tightened by (Niu et al., 2024) to the order $\Omega\left(\sqrt{\frac{d}{N\lambda_k}} + \sqrt{\frac{Md}{N^2\lambda_k^2}}\right)$.

For the well-represented case, the upper and lower bounds in (Tripuraneni et al., 2021) are $O(\sqrt{\frac{dk^2}{N}})$ and $\Omega(\sqrt{\frac{dk}{N}})$, respectively, resulting in a multiplicative gap of $\Theta(\sqrt{k})$. Meanwhile, the estimator proposed in (Niu et al., 2024) achieves a rate of $O\left(\sqrt{\frac{dk}{N}} + \sqrt{\frac{Mdk^2}{N^2}}\right)$, which matches the state-of-the-art minimax lower bound. In other words, for the well-represented case, the estimator in (Niu et al., 2024) is statistically minimax optimal.

Table 1 summarizes the state-of-the-art (SOTA) bounds in both general and well-represented cases. A more detailed comparison of upper and lower bounds across recent works is presented in Table 3 in the Appendix.

*Table 1.* Statistical error rate depends on the spectrum of $D$. Here, UB stands for Upper Bound, and LB stands for Lower Bound.

| | | **SOTA (Niu et al., 2024)** |
|---|---|---|
| **General Cases** | UB | $O\left(\sqrt{\frac{d\lambda_1}{N\lambda_k^2}} + \sqrt{\frac{Md}{N^2\lambda_k^2}}\right)$ |
| | LB | $\Omega\left(\sqrt{\frac{d}{N\lambda_k}} + \sqrt{\frac{Md}{N^2\lambda_k^2}}\right)$ |
| **Well-represented Cases** | UB | $\Theta\left(\sqrt{\frac{dk}{N}} + \sqrt{\frac{Mdk^2}{N^2}}\right)$ |
| | LB | |

Throughout this paper, we use the Bachmann–Landau notations $o, \omega, O, \Omega$, and $\Theta$, and use $\widetilde{O}$ to hide polylogarithmic factors in quantities.

# 4. Main Results

This section is structured as follows: we first present (in Section 4.1) a special motivating example where the local heads are mutually orthogonal to show the benefit of the source screening. Based on this motivating example, Section 4.2 formalizes what constitutes a statistically desirable subpopulation through the notion of an admissible subpopulation, namely, a subset of sources whose aggregate local-head geometry is well-conditioned and whose size is sufficiently large. Section 4.3 is devoted to the design of algorithms for identifying admissible subpopulations: Algorithm 1 gives a genie-aided search method, and Algorithm 2 gives an empirical version based on data-driven proxies.

## 4.1. Potentials of Source Screening

In this section, through theoretical derivation and numerical illustration, we provide insights into the potential of source screening to improve statistical rates – despite discarding a large portion of the data. We study a special yet practically important family of problem instances in which there are $k$ distinct local heads that are mutually orthogonal and unit normed, i.e., letting $\nu_{[1]}^\star, \ldots, \nu_{[k]}^\star$ denote the $k$ distinct local heads, we assume $\langle \nu_{[i]}^\star, \nu_{[j]}^\star \rangle = 0$ for all $i \neq j$, and $\|\nu_{[i]}^\star\| = 1$ for all $i \in [k]$. We work in the balanced-sample regime $n_i = n$ for all $i \in [M]$, so $N = Mn$.

This orthogonal and normalized head setting provides both conceptual and analytical clarity. Orthogonality isolates the role of source composition from interactions induced by overlapping local structures. Moreover, the eigensystem of $D$ are explicitly computable, with eigenvalues and eigenvectors naturally linked to the diversity of local heads and the

associated data volumes of different sources. This transparent characterization enables a fine-grained understanding of how source composition affects subspace learning and when source screening becomes beneficial.

Let $m_j$ denote the total data volume along $\nu_{[j]}^\star$, i.e., $m_j = n \sum_{i=1}^M \mathbf{1}_{\left\{\alpha_i^\star = \nu_{[j]}^\star\right\}}$. It is easy to see that $\sum_{j=1}^k m_j = N$. Without loss of generality, let $m_1 \geq m_2 \geq \cdots \geq m_k$. The diversity matrix $D$ can be rewritten as

$$D = \frac{1}{N} \sum_{j=1}^k m_j \nu_{[j]}^\star (\nu_{[j]}^\star)^\top = \sum_{j=1}^k \frac{m_j}{N} \nu_{[j]}^\star (\nu_{[j]}^\star)^\top.$$

It is easy to see that $\nu_{[j]}^\star$ for $j \in [k]$ act as eigenvectors of the matrix $D$. The corresponding eigenvalues are

$$\lambda_1 = \frac{m_1}{N} := \beta_1, \lambda_2 = \frac{m_2}{N} := \beta_2, \cdots, \lambda_k = \frac{m_k}{N} := \beta_k.$$

Plugging those $\lambda_j$'s into the upper bound in Table 1, we know that using the SOTA subspace estimator in (Niu et al., 2024), one can achieve

$$\|\sin\Theta(B, B^\star)\| = O\left(\sqrt{\frac{d}{N}\frac{\beta_1}{\beta_k^2}} + \sqrt{\frac{Md}{N^2\beta_k^2}}\right). \quad (5)$$

Now consider the genie-aided scenario in which, for each $j \in [k]$, exactly $m_k/n$ sources are retained, while the remaining sources are discarded. Let $\mathcal{S}$ denote the collection of retained sources. Clearly, $|\mathcal{S}| = km_k/n = k\beta_k M$. Denote the new data diversity matrix as $D'$, i.e.,

$$D' = \frac{1}{\sum_{j=1}^k m_k} \sum_{j=1}^k m_k \nu_{[j]}^\star (\nu_{[j]}^\star)^\top = \frac{1}{k} \sum_{j=1}^k \nu_{[j]}^\star (\nu_{[j]}^\star)^\top.$$

It is easy to see that $\nu_{[j]}^\star$'s remain to be the eigenvectors of $D'$ but with uniform eigenvalues, i.e., $\lambda_j' = \frac{1}{k}$ for $j \in [k]$. Let $N' = km_k$ denote the total data volume in this scenario. Notably, when restricted to the subpopulation of sources $\mathcal{S}$, the upper bound in Table 1 continues to hold. We have

$$\|\sin\Theta(B, B^\star)\| = O\left(\sqrt{\frac{d}{N\beta_k}} + \sqrt{\frac{|\mathcal{S}|d}{N^2\beta_k^2}}\right), \quad (6)$$

whose first term is tighter than that of Eq. (5) by a factor of $\frac{\beta_1}{\beta_k}$, which can be significant as $\frac{\beta_1}{\beta_k} \gg 1$ is possible. The second term is also tighter, scaling with the number of sources $|\mathcal{S}|$ that remain in the system.

On the other hand, a lower bound in (Niu et al., 2024) (particularly, Theorem 5.1) holds for these particular choices of local heads.

**Theorem 1** ((Niu et al., 2024) (Informal)). *Consider a system with $M$ sources and $N$ data points in total. Assume*

$x_{ij} \sim N(0, I_d)$ *and* $\xi_{ij} \sim N(0,1)$ *independently for* $i \in [M]$ *and* $j \in [n]$. *Then for the model in Eq. (2), when* $d \geq (1 + \rho_1)k$ *for a constant* $\rho_1 > 0$, *we have*

$$\inf_{\widehat{B} \in \mathcal{O}^{d \times k}} \sup_{B \in \mathcal{O}^{d \times k}} \mathbb{E}\left[\|\sin\Theta(\widehat{B}, B)\|\right] = \Omega\left(\sqrt{\frac{d}{N\lambda_k}} \wedge 1\right),$$

*where* $\sqrt{\frac{d}{N\lambda_k}} \wedge 1 = \min\{\frac{d}{N\lambda_k}, 1\}$, *and* $\mathcal{O}^{d \times k}$ *is the collection of orthonormal matrices as per Definition 1.*

When $|\mathcal{S}| \leq M \leq N\beta_k = m_k$, Eq. (6) can be simplified as

$$\|\sin\Theta(B, B^\star)\| = O\left(\sqrt{d/(N\beta_k)}\right),$$

matching the lower bound in Theorem 1. In other words, for this family of problem instances on local heads $\{\alpha_i^\star\}_{i=1}^M$, in the genie-aided case, when a good subset of sources is given, the SOTA subspace estimator (i.e., the local averaging method) in (Niu et al., 2024) is **statistically minimax optimal**, despite a large portion of sources being discarded.

Beyond our theoretical results, we numerically illustrate the performance gap between naive data pooling and strategic subset selection in Fig.1. In settings where certain source groups dominate the population, standard estimators often fail to recover the shared subspace accurately due to the resulting representational bias. We show that by intentionally downsampling the majority groups to achieve a balanced distribution of heads, one can drastically reduce the reconstruction error (Definition 1) despite utilizing fewer total samples. This counterintuitive result demonstrates that for subspace learning, the overall composition of the data sources are often more critical than the raw sample volume. A comprehensive description of the estimators and the specific data-generating process is provided in Section 6.

## 4.2. On the Fundamentals of Source Screening

We now move beyond the orthonormal construction used in Section 4.1 and ask a more general question: what property should a subpopulation of sources satisfy in order to achieve statistical minimax optimality?

### 4.2.1. DESIRED SUBPOPULATION AND EXISTENCE

We begin with a couple of concrete examples to build intuition and motivate a formal definition of a desired subpopulation of sources. Recall from the motivating example in Section 4.1 that $\beta_1 = \frac{m_1}{N}$ and $\beta_k = \frac{m_k}{N}$, i.e., they are the fractions of sources whose underlying truth is $\nu_{[1]}^\star$ and $\nu_{[k]}^\star$, respectively. Let $\mathcal{S}_\ell \subseteq \{i : \alpha_i^\star = \nu_{[\ell]}^\star\}$ such that $|\mathcal{S}_\ell| = \beta_k M$ for $\ell = 1, \cdots, k$. Those subsets exist because $|\{i : \alpha_i^\star = \nu_{[\ell]}^\star\}| \geq |\{i : \alpha_i^\star = \nu_{[k]}^\star\}| = \beta_k M$. Let $\mathcal{S} = \cup_{\ell=1}^k \mathcal{S}_\ell$, and $A = [\alpha_1^\star, \cdots, \alpha_M^\star]$. It is easy to verify that the condition number, denoted by $\kappa(\cdot)$, of the matrix $\sum_{i \in \mathcal{S}} \alpha_i^\star(\alpha_i^\star)^\top$ is

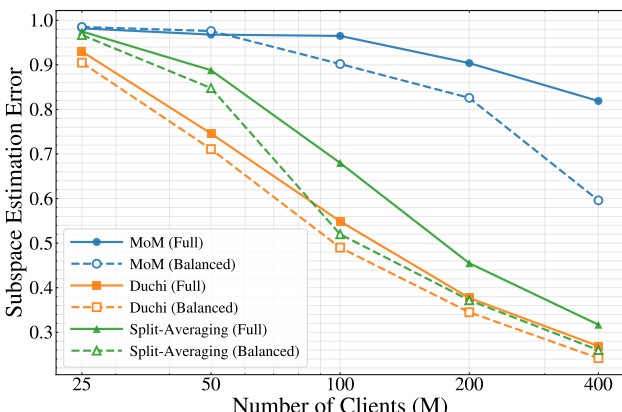

*Figure 1.* Subspace estimation error ($\|\sin(B^\star, B)\|$) in a clustered setting. While pooling the full population maximizes sample size, uneven representation introduces bias. Conversely, a smaller balanced subset recovers the latent basis more effectively across all tested estimators. See Section 6 for setup and estimator details.

at a constant level, i.e., $\kappa\left(\sum_{i \in \mathcal{S}} \alpha_i^\star(\alpha_i^\star)^\top\right) = \Theta(1)$. In addition, $|\mathcal{S}| = k\beta_k M = k\lambda_{\min}(AA^\top)$, where $\lambda_{\min}(AA^\top)$ represents the smallest eigenvalue of $AA^\top$. From our analysis in Section 4.1, we know that training on the data kept on this subpopulation of sources with the split-local averaging algorithm in (Niu et al., 2024) is minimax optimal.

At the opposite extreme, when $\alpha_i^\star \overset{i.i.d.}{\sim} \mathcal{N}(\mathbf{0}, \mathbf{I})$, it follows from (Niu et al., 2024) that, with high probability, $\kappa\left(\sum_{i \in [M]} \alpha_i^\star(\alpha_i^\star)^\top\right) = \Theta(1)$, and $|\mathcal{S}| = \Theta\left(k\lambda_{\min}(AA^\top)\right) = \Theta\left(k\frac{M}{k}\right) = \Theta(M)$. Furthermore, training on the data kept on all sources achieves statistical minimax optimality.

Building on these insights, we provide a formal notion of a desired subpopulation as follows. As a gentle entry point to this line of investigation, yet without losing much generality, we assume matrix $A$ is standardized, i.e., each column has norm 1 (Tropp, 2009).

**Definition 2** (Admissible subpopulation). *Let* $A \in \mathbb{R}^{k \times M}$ *be a standardized matrix such that its columns span* $\mathbb{R}^k$. *We say a subpopulation* $\mathcal{S} \subseteq [M]$ *is admissible if* $\kappa\left(\sum_{i \in \mathcal{S}} \alpha_i^\star(\alpha_i^\star)^\top\right) = \Theta(1)$, *and* $|\mathcal{S}| = \Theta(k\lambda_{\min}(AA^\top))$.

**Theorem 2** (Statistical minimax optimality). *Let* $\mathcal{S} \subseteq [M]$ *be a given subpopulation that satisfies the conditions in Definition 2. Suppose that each source independently collects local datasets that satisfy Assumptions 2 and 1. Suppose that* $n_i = n$ *for* $i \in [M]$. *When* $\lambda_k \geq \frac{1}{n}$, *one can achieve minimax optimal rates* $O(\sqrt{d/(N\lambda_k)})$ *by restricting training to the datasets on this subpopulation only.*

Intuitively, this result shows that as long as there exists a sufficiently "good" subpopulation of sources, training exclusively on this group is already statistically optimal,

despite disregarding the data on other sources. In other words, restricting attention to such a carefully chosen subset does not incur any statistical loss; rather, it can lead to a significant statistical gain by mitigating bias introduced by less informative or misaligned sources.

It is worth noting that, in the proof of Theorem 2, the optimal rate is achieved using the split-sample averaging algorithm proposed in (Niu et al., 2024). However, our results are not tied to this specific algorithm. In fact, any algorithm that attains statistical minimax optimality under well-conditioned settings would suffice.

When $\lambda_{\min}(AA^\top) = O(1)$, a good subpopulation as per Definition 2 often exists, formally stated in Theorem 3.

**Theorem 3.** *Let $A = [\alpha_1^\star, \cdots, \alpha_M^\star] \in \mathbb{R}^{k \times M}$ be standardized. Suppose that $\|A\|^2 \lesssim \frac{M}{k}$. When $M$ is sufficiently large, there exists a subset $\mathcal{S} \subseteq [M]$ such that*

$$\kappa\big(\textstyle\sum_{i \in \mathcal{S}} \alpha_i^\star (\alpha_i^\star)^\top\big) = \Theta(1), \text{ and } |\mathcal{S}| = \Omega(k\lambda_{\min}(AA^\top)).$$

It is worth noting that the condition $\|A\|^2 \lesssim \frac{M}{k}$ does not imply the condition number of $A$ is small. In fact, its condition number can be arbitrarily bad. To see this, consider the example in which $k$ is even, and the top $k/2$ eigenvalues of $AA^\top$ are all $\left(\frac{2M}{k} - \epsilon\right)$ for some small $\epsilon$, while the remaining $k/2$ eigenvalues are all $\epsilon$. In this example, $\kappa(A) = \sqrt{\frac{\frac{2M}{k} - \epsilon}{\epsilon}}$, which can be arbitrarily large. Theorem 3 indicates that even if the full matrix $A$ is poorly conditioned, one can still identify a sufficiently large subset of sources whose collective structure is well-conditioned. Conversely, including all sources may be detrimental to the training.

The proof of Theorem 3 builds upon the notion of stable rank and the classic result in (Bourgain & Tzafriri, 1987).

The stable rank of a matrix is defined as

$$\text{st.rank}(A) := \|A\|_F^2 / \|A\|^2. \tag{7}$$

Intuitively, the stable rank measures the effective "energy spread" of a matrix. Unlike the traditional rank, it tells you how many "significant directions" the matrix really has, in a way that is robust to small perturbations. In general, $\text{st.rank}(A) \leq \text{rank}(A)$.

Let $A_{\mathcal{S}_0}$, where $\mathcal{S}_0 \subseteq [M]$, denote the submatrix of $A$ that contains only the collection of columns in $\mathcal{S}_0$.

**Theorem 4.** *(Bourgain & Tzafriri, 1987) Suppose matrix $A$ is standardized, and $\text{st.rank}(A) = \omega(1)$. Then for sufficiently large $M$ and $k$, there exists a set $\mathcal{S}$ of columns such that $|\mathcal{S}| = \lceil c \cdot \text{st.rank}(A) \rceil$ and $\kappa\left(A_{\mathcal{S}} A_{\mathcal{S}}^\top\right) \leq 3$, for some absolute constant $c > 0$.*

We prove Theorem 4 for completeness in the Appendix I.

### 4.2.2. ALGORITHM IN THE GENIE-AIDED SELECTION

Motivated by the encouraging findings in Section 4.2.1, we next turn to algorithmic solutions. Fortunately, the proof of Theorem 3 is partially constructive, while the analysis in Theorem 4 provides additional algorithmic insights. Together, they naturally lead to a polynomial-time algorithm for identifying a desired subpopulation.

---

**Algorithm 1** Genie-aided Subpopulation Search

---

1: **Input:** A full-row-rank standardized matrix $A = [\alpha_1^\star, \alpha_2^\star, \cdots, \alpha_M^\star]$, where $\|\alpha_i^\star\|_2 = 1$ for $i = 1, \cdots, M$, and $\|A\|^2 \lesssim \frac{M}{k}$. An absolute constant $c^*$ for which $320(c^* + \sqrt{2c^*}) \leq 0.5$. A target success rate $\delta \in (0, 1)$;

2: **Output:** A set of column indices $\mathcal{S} \subseteq [M]$.

3: Compute st.rank($A$).

4: **if** $c^* \cdot \text{st.rank}(A) < 1$ **then**

5:      **Return** $\emptyset$, and display *"low stable rank"*;

6: **end if**

7: $s \leftarrow \lceil c^* \cdot \text{st.rank}(A) \rceil$, $A_1 \leftarrow A$, and $\mathcal{S} \leftarrow \emptyset$;

8: Compute $\lambda_{\min}(AA^\top)$;

9: Compute $\|A_1\|^2$;

10: $\widetilde{c} \leftarrow \frac{\|A_1\|^2}{M/k}$;

11: **for** $t = 1, \cdots, \lceil \lambda_{\min}(AA^\top) \rceil$ **do**

12:      **if** st.rank($A_t$) $\geq \frac{k}{2\widetilde{c}}$ **then**

13:          **for** $\ell = 1, ..., \log_{8/7} \frac{\lambda_{\min}(AA^\top)}{\delta}$ **do**

14:              Draw uniformly at random $\widetilde{\mathcal{S}}_t$ with size $s$;

15:              Compute the $H_{\widetilde{\mathcal{S}}_t \times \widetilde{\mathcal{S}}_t} = A_{\widetilde{\mathcal{S}}_t}^\top A_{\widetilde{\mathcal{S}}_t} - I_s$;

16:              **if** $\|H_{\widetilde{\mathcal{S}}_t \times \widetilde{\mathcal{S}}_t}\|_{\infty \to 1} \leq \frac{s}{8}$ **then**

17:                  Perform Grothendieck Factorization on $H_{\widetilde{\mathcal{S}}_t \times \widetilde{\mathcal{S}}_t}$ to obtain $H_{\widetilde{\mathcal{S}}_t \times \widetilde{\mathcal{S}}_t} = D_t T_t D_t$;

18:                  Let $\mathcal{S}_t = \{j : d_{jt}^2 \leq 2/s, j \in \widetilde{\mathcal{S}}_t\}$, where $d_{jt}$ is the $j$-th diagonal entry of $D_t$;

19:                  **Break**.

20:              **end if**

21:          **end for**

22:          Remove columns in $\mathcal{S}_t$ from $A_t$ to obtain $A_{t+1}$;

23:      **else**

24:          **Break**

25:      **end if**

26: **end for**

27: Set $t^* \leftarrow t$;

28: **if** st.rank($A_{t^*}$) $\geq \frac{k}{2\widetilde{c}}$ **then**

29:      $\mathcal{S} \leftarrow \cup_{r=1}^{t^*} \mathcal{S}_r$;

30: **else**

31:      $\mathcal{S} \leftarrow \cup_{r=1}^{t^*-1} \mathcal{S}_r$;

32: **end if**

---

The resulting procedure is summarized in Algorithm 1. Its outer loop follows the constructive selection scheme be-

hind Theorem 3: at round $t$, it searches within the current residual matrix $A_t$ for a block $\mathcal{S}_t$, removes the selected columns to form $A_{t+1}$, and repeats while the residual matrix has sufficient stable rank. The inner loop carries out this search through repeated random trials, as suggested by Theorem 4: it samples a candidate set $\widetilde{\mathcal{S}}_t$, checks whether the candidate columns have a Gram matrix close to the identity, i.e., whether $A_{\widetilde{\mathcal{S}}_t}^\top A_{\widetilde{\mathcal{S}}_t} - I_s$ is small in $\|\cdot\|_{\infty\to 1}$, and then uses Grothendieck factorization to prune $\widetilde{\mathcal{S}}_t$ into a well-conditioned block $\mathcal{S}_t$. The final output is the union of all accepted blocks.

**Theorem 5.** *Let $c^*$ denote an absolute constant such that $320(c^* + \sqrt{2c^*}) \leq 0.5$. For any given $\delta \in (0,1)$, with probability at least $1-\delta$, Algorithm 1 outputs an admissible subset of sources $\mathcal{S}$ as per Definition 2.*

### 4.3. Empirical Subpopulation Search

Algorithm 1 relies on (1) the stable rank of the inaccessible matrix $A$ and (2) $\lambda_{\min}(AA^\top)$ in determining the size of the randomly selected columns. In practice, $A$ is not given. In this section, we present practical heuristics for algorithm design that circumvent the need for this information.

For simplicity, we assume $n_i = n, \forall i \in [M]$, with $n$ even. Let $\bar{z}_i = \frac{2}{n_i} \sum_{j=1}^{n_i/2} y_{ij} x_{ij}$, and $\widetilde{z}_i = \frac{2}{n_i} \sum_{j=n_i/2+1}^{n_i} y_{ij} x_{ij}$. Let $Z = \frac{1}{N} \sum_{i=1}^M n_i \bar{z}_i \widetilde{z}_i^\top$. Next, we discuss the connection between matrix $A$, $B^\star A$, $D$, and $Z$. We know $\mathbb{E}[Z] = B^\star D(B^\star)^\top$. Note that the nonzero eigenvalues of $\mathbb{E}[Z]$ are identical to those of matrix $D$. Formally,

$$\lambda_{\min}(D) = \lambda_{\min}^+(B^\star D(B^\star)^\top) = \lambda_{\min}^+(\mathbb{E}[Z]),$$

where $\lambda_{\min}^+$ denotes the smallest non-zero eigenvalue. In addition, the eigenvectors of $\mathbb{E}[Z]$ are $\{B^\star v_\ell\}_{\ell=1}^k$, where $\{v_\ell\}_{\ell=1}^k$ are the eigenvectors of $D$.

Note that if we define the scaled version $\widetilde{A} = \frac{1}{\sqrt{N}}[\sqrt{n}\alpha_1^\star, \cdots, \sqrt{n}\alpha_M^\star]$, we have $D = \widetilde{A}\widetilde{A}^\top$, and

$$\lambda_{\min}\left(\widetilde{A}\widetilde{A}^\top\right) = \lambda_{\min}^+(\mathbb{E}[Z]).$$

Despite this connection, we are still not able to directly work on $B^\star\widetilde{A}$ as Algorithm 1 requires the unscaled matrix $A$. Instead, we work with the unscaled version of the matrix $Z$, i.e., $\sum_{i=1}^M \bar{z}_i \widetilde{z}_i^\top$. Let $\widehat{Z} = \sum_{i=1}^M \bar{z}_i \widetilde{z}_i^\top$. Then,

$$\mathbb{E}[\widehat{Z}] = \sum_{i=1}^M \mathbb{E}[\bar{z}_i]\mathbb{E}[\widetilde{z}_i^\top] = B^\star AA^\top(B^\star)^\top.$$

By the same logic as in the scaled case, the nonzero eigenvalues of $\mathbb{E}[\widehat{Z}]$ are identical to those of the matrix $AA^\top$. Thus, we have $\lambda_{\min}(AA^\top) = \lambda_{\min}^+(B^\star AA^\top(B^\star)^\top) = \lambda_{\min}^+\left(\mathbb{E}[\widehat{Z}]\right)$. In addition, we can prove st.rank$(B^\star A) =$

st.rank$(A)$. The proof is deferred to Section K.1. Then, we can develop an empirical version of Algorithm 1, which replaces $A$ by $(\bar{Z}, \widetilde{Z})$. Under standard concentration assumptions on the local sample size $n$, $\widehat{Z}$ concentrates around its expectation in the operator norm, and st.rank$(\bar{Z}+\widetilde{Z})$ serves as a consistent proxy for st.rank$(A)$. The resulting empirical procedure is formally stated in Algorithm 2, which can be found in Appendix E.

## 5. Extension to Nonlinear Models

In this section, we show that the same source-screening argument extends to nonlinear models, as the screening step depends on the geometry of the local-head matrix rather than on the linearity of the response model.

Consider the nonlinear model

$$\mathbb{E}[y_{ij} \mid x_{ij}] = h_i\big((B^\star)^\top x_{ij}\big), \tag{8}$$

where $B^\star \in \mathbb{R}^{d\times k}$ has orthonormal columns and $h_i : \mathbb{R}^k \to \mathbb{R}$ is a source-specific nonlinear function. Throughout this section, we follow the nonlinear setup of Niu et al. (2024, Section 7): the covariates satisfy $x_{ij} \sim N(0, I_d)$ independently across $i \in [M]$ and $j \in [n]$, and the functions $h_i$ satisfy the regularity assumptions imposed there. For completeness, we provide the assumptions in Appendix C.

Let $U \sim N(0, I_k)$. Define the effective local head of source $i$ as $\alpha_i^\star := \mathbb{E}_{U\sim N(0,I_k)}[h_i(U)U] \in \mathbb{R}^k$. Lemma 7.1 of Niu et al. (2024) shows that the source-level quantities used by the split-averaging estimator have expectation $B^\star\alpha_i^\star$. Intuitively, $\alpha_i^\star$ plays the role of the local head in the nonlinear model. With these effective heads replacing the linear local heads, we reuse the notation

$$A = [\alpha_1^\star, \ldots, \alpha_M^\star] \in \mathbb{R}^{k\times M}, \qquad D = \frac{1}{M}AA^\top.$$

Let $\lambda_1 \geq \cdots \geq \lambda_k > 0$ denote the eigenvalues of $D$. Here $D$ is the effective source diversity matrix.

As in the linear analysis, we work in the standardized case where $\|\alpha_i^\star\|_2 = 1$ for all $i \in [M]$, and assume the columns of $A$ span $\mathbb{R}^k$. A subpopulation $S \subseteq [M]$ is admissible for the nonlinear model if it satisfies Definition 2 with respect to the effective local heads above.

This formulation reduces nonlinear source screening to the same matrix-geometric problem studied in the linear setting. The existence and genie-aided search results depend only on the matrix $A$, its stable rank, and the conditioning of selected column submatrices; they do not depend on the specific functional form of $h_i$.

It remains to connect this geometric reduction to the statistical rate. Under the assumptions in Appendix C, the nonlinear split-averaging result of Niu et al. (2024, Theorem 7.1) has the same spectral dependence on the diversity

matrix formed from the effective local heads. Since the source-screening argument in Theorem 2 depends only on this diversity matrix and the admissibility of $S$, the same argument applies after replacing the linear local heads by the effective local heads. Thus, under the same conditions as in Theorem 2, training on an admissible subpopulation attains the same minimax-optimal rate.

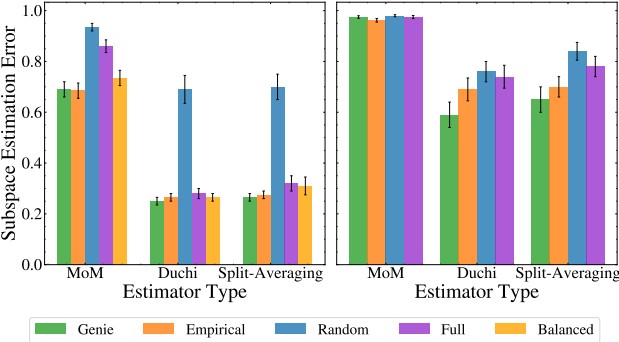

*Figure 2.* (Left) Performance on Clustered $\alpha_i$ setting. (Right) Performance on Heterogeneous Gaussian $\alpha_i$ setting.

## 6. Numerical Experiments

We validate our proposed data source pre-screening method through experiments on synthetic regression tasks and real-world classification benchmarks. While our main results study the linear representation learning case, we provide additional ablations, such as nonlinear feature extractors, in Appendix D. Overall, these evaluations demonstrate the resilience of our approach across diverse data regimes (e.g., dimensionality, correlation structures) and system scales.

We compare our method against several baselines: *full population training* (no screening), *random subsampling*, and *power-of-choice* selection (Cho et al., 2020) (applied after one initial round of full population training). To ensure parity, we constrain all baselines to select the same number of sources as our screening procedure. In the clustered synthetic setting, we additionally evaluate a *balanced* baseline that subsamples sources using ground-truth cluster assignments to guarantee equal group representation. For all algorithms, downstream estimator performance is averaged over 20 random seeds.

### 6.1. Synthetic Data

In the synthetic setting, we evaluate our genie-aided (1) and empirical (2) algorithms within a linear regression framework. After performing source selection, we measure the performance of the resulting subpopulation using three representative estimators: *Median-of-Means* (MoM) (Tripuraneni et al., 2021), which offers algorithmic simplicity while reaching sharp statistical limits under moderate heterogeneity; *DFHT* (Duchi et al., 2022), with strong em-

pirical performance and robust theoretical guarantees; and the *split-averaging estimator* (Niu et al., 2024), which is provably minimax optimal in balanced settings. Following existing literature, our metric in this setting is the principal angle distance between the joint subspace estimate $B$ and the underlying truth $B^\star$: $\|\sin\Theta(B^\star, B)\|$.

**Synthetic Data Generation.** We evaluate our proposed methods using synthetic distributed data generated under a source-specific linear model: $y_{ij} = x_{ij}^\top B^\star \alpha_i^\star + \epsilon_{ij}$, where $B^\star$ represents a shared subspace and $\alpha_i^\star$ are source-specific coefficients. To necessitate robust collaborative learning, local sample sizes $n_i$ are deliberately restricted ($n_i \leq d$) so that purely local subspace recovery is insufficient.

To assess the robustness of our subspace estimation, we consider two distinct regimes for the coefficients $\alpha_i^\star$:

- **Clustered Coefficients:** Sources are probabilistically assigned to one of two groups, with each group occupying a disparate half of the latent subspace. Specifically, a source belongs to the first group with probability $g$ (default $g = 0.2$), where its coefficients have non-zero variance only on the first $k/2$ dimensions.

- **Heterogeneous Gaussian:** Coefficients are drawn from zero-mean Gaussian distributions $\alpha_i \sim \mathcal{N}(0, \Psi_i)$, where each $\Psi_i$ is a randomly generated positive semi-definite (PSD) covariance matrix.

Full details regarding data generation, standard parameter values (e.g., $M$, $d$, $k$), and covariance construction are provided in Appendix D.1.

**Results.** The results for subspace estimation under the clustered and heterogeneous Gaussian regimes are illustrated in Fig. 2. In both settings, our empirical algorithm (Algorithm 2) consistently achieves a lower subspace reconstruction error than using the full source population. This gap is particularly pronounced in the clustered setting, suggesting that our selection mechanism effectively identifies the most informative sources for the shared basis even when population heads are sparse. Furthermore, the performance of the balanced and genie-aided algorithms indicates the potential performance under ideal prescreening conditions.

In Fig. 3, we examine the impact of problem dimensionality on subspace recovery. While increasing $d$ and $k$ inherently raises the task complexity, our proposed methods strictly outperform training on the full source population. Notably, the genie-aided algorithm exhibits superior resilience to increases in the latent rank $k$ compared to existing estimators. In Fig. 4, we examine the effects of the source count and distribution. Our method is able to identify subpopulations superior to full-population training even when $M$ is small, with greater benefits in more imbalanced settings.

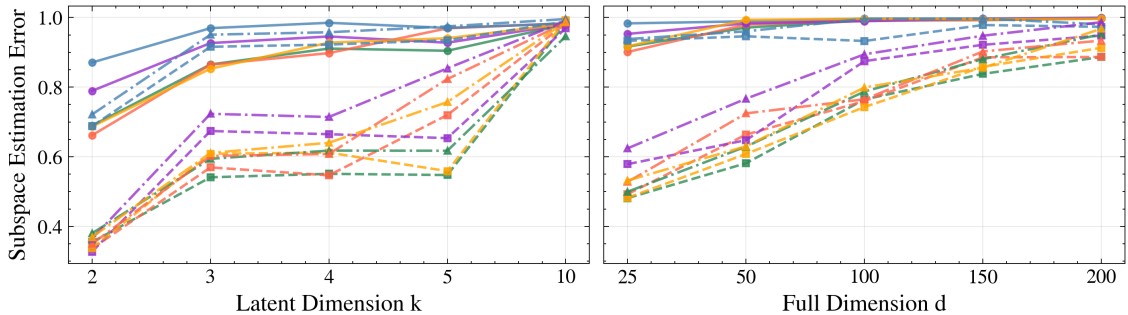

*Figure 3.* (Left) Ablation over latent dimensionality $k$. (Right) Ablation over full dimensionality.

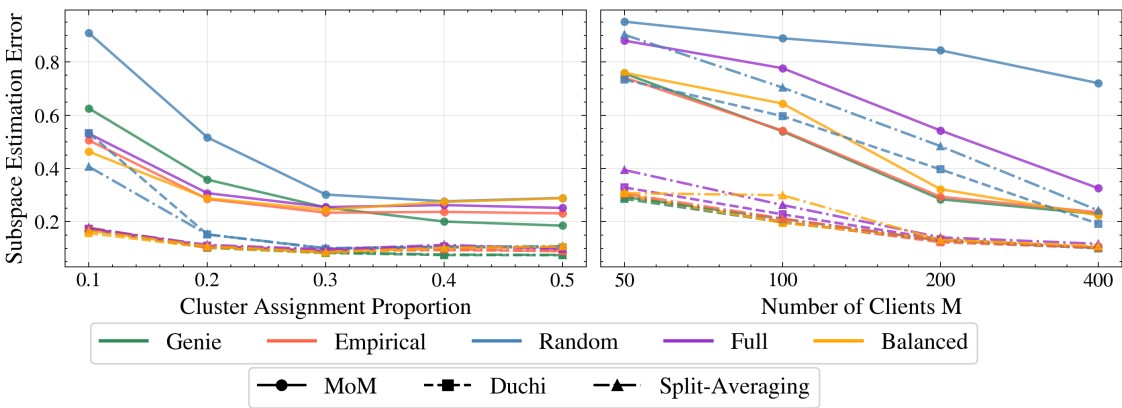

*Figure 4.* (Left) Ablation over clustered sources assignment proportion. (Right) Ablation over the number of sources $M$.

## 6.2. Real-world Data.

To assess practical applicability, we evaluate our method on the ACSIncome (Ding et al., 2021) and CelebA (Liu et al., 2015) datasets. These benchmarks provide natural data partitions that reflect real-world distribution shifts and allow us to test our method across regimes of varying complexity. For all tasks, we adopt `FedRep` (Collins et al., 2021) as the base estimator, run our algorithm with varying assumed values of dimension $k$, and report the final classification accuracy averaged over the entire source population.

**ACSIncome.** The ACSIncome dataset is used to predict whether an individual's annual income exceeds $50,000. We partition the approximately 1.6 million low-dimensional samples ($d = 10$) geographically by state to simulate natural heterogeneity.

**CelebA.** We utilize the CelebA dataset for smile classification. Following the LEAF benchmark (Caldas et al., 2018), we select a 10% subset partitioned by individual identity. Given the high dimensionality of the images, we employ a pre-trained Vision Transformer (ViT-Tiny) (Dosovitskiy, 2020) to extract a lower-dimension signal ($d = 192$) prior to subset selection and fine-tuning.

Additional details on the dataset configuration and preprocessing are provided in Appendix D.2.

**Results.** Table 2 summarizes the classification performance on real-world datasets. Our method demonstrates a clear advantage over both uniform and active sampling baselines.

|  | ACSIncome (%) | CelebA (%) |
|---|---|---|
| Full Population | 72.8 | 89.5 |
| Random Selection | 71.2 | 88.3 |
| Power-of-Choice | 73.0 | 89.8 |
| Ours ($k = 3$) | 73.4 | 90.0 |
| Ours ($k = 5$) | 73.9 | 90.3 |
| Ours ($k = 7$) | 74.2 | 90.5 |

*Table 2.* Binary Classification Accuracy on Income and CelebA.

## 7. Conclusion

We studied the problem of learning shared linear subspaces from heterogeneous data sources. Our theoretical analysis establishes that, under mild regularity conditions, a balanced subcollection of sources can achieve minimax optimal rates and outperform full data utilization. Our experiments confirm that our method consistently improves estimation accuracy across both linear and non-linear regimes. A promising direction for future work involves extending our theoretical analysis in empirical settings.

## Impact Statement

We do not anticipate direct negative societal or ethical consequences. While the practical impact of data pre-screening depends on the application context, our results suggest that thoughtful source selection can, in fact, promote more equitable learning outcomes by counteracting imbalance rather than reinforcing it. There are many other potential societal consequences of our work, none which we feel must be specifically highlighted here.

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

# Appendices

## A. Related Work on Source Selection in Federated Learning

In federated learning, a parameter learner focuses on training a global model or personalized model via minimizing an objective of the form in Eq. (1). Eq. (1) itself is a classic personalized federated learning formulation. When a global model is trained, $\Phi = \{\mathcal{I}\}$ is the identity mapping, and local heads are forced to be the same across sources, i.e., $h_i = h_{i'}$ for $i, i'$ (Kairouz et al., 2021).

Partial source participation has been widely adopted in FL to mitigate communication costs. The selection scheme has been developed in recent years. (Li et al., 2019), (Yang et al., 2021), (Karimireddy et al., 2019), (Qu et al., 2021) treat sources equally important, and the sources are sampled uniformly at random. (Li et al., 2020) selects sources according to the ratio of their local data volume and the total data volume. However, when the data are heterogeneous, these sampling schemes suffer from slow convergence due to high variance. (Luo et al., 2022) proposes an adaptive source sampling algorithm that tackles heterogeneity to minimize convergence time. (Chen et al., 2020) utilizes importance sampling and proposes an adaptive source sampling strategy to reduce communication bandwidth. (Fraboni et al., 2021) introduces clustered sampling based on sources' sample size or model similarity, which consistently leads to better convergence performance. (Cho et al., 2020) selects sources with higher training loss, resulting in an increase in convergence rate. (Ruan et al., 2024) builds a comprehensive system for evaluating the source selection strategy based on quantitative metrics, including training loss, generalization error, population representativeness, completion time, and differential privacy. (Xiang et al., 2024) studies non-stationary source availability and proposes a novel algorithm that compensates for missed information due to source unavailability, although the non-stationary dynamics are unknown.

## B. Standard Assumptions and Statistical Rates of Existing Work

We impose the following assumptions on the noise variables $\xi_{ij}$, the covariate vectors $x_{ij}$, and the parameters $\theta_i^\star$.

**Assumption 1** (Sub-Gaussian noises). The noise variables $\xi_{ij}$ are independent, zero-mean, sub-Gaussian[1] with constant variance proxy $\sigma^2 = \Theta(1)$ and are independent of covariates $x_{ij}$.

**Assumption 2** (Sub-Gaussian covariates). The covariates $x_{ij}$ are independent, zero-mean, sub-Gaussian with variance proxy $\gamma^2 = \Theta(1)$. For each $i$, $x_{ij}$ share the same but *unknown* covariance, i.e., $\mathbb{E}[x_{ij}x_{ij}^\mathsf{T}] = \Gamma_i$ for all $j$. These covariance matrices are well-conditioned, with $\lambda_1(\Gamma_i)/\lambda_d(\Gamma_i) = \Theta(1)$ for all $i$.

The sub-Gaussian assumptions are standard in statistical learning for deriving tail bounds. Assumption 2 generalizes those in (Tripuraneni et al., 2021; Duchi et al., 2022) by allowing non-identity covariance.

**Assumption 3** (Source normalization). Each $\alpha_i^\star$ satisfies $\|\alpha_i^\star\| = O(1)$ for $i \in [M]$.

The normalization assumption is standard in the literature. Recall that $\lambda_r = \lambda_r(D)$ is the $r$-th largest eigenvalue of the source diversity matrix $D$ defined in Eq. (4) for $r \in [k]$. The normalization then gives $\sum_{r=1}^k \lambda_r = \text{trace}(D) = \text{trace}(\sum_{i=1}^M n\alpha_i^\star(\alpha_i^\star)^\mathsf{T})/N = \sum_{i=1}^M n\|\alpha_i^\star\|^2/N = O(1)$, which further implies that $k\lambda_k \leq O(1)$ and $\lambda_1 = O(1)$.

## C. Assumptions for Nonlinear Models

**Assumption 4.** The covariates $x_{ij}$ are sampled i.i.d. from $N(0, I_d)$ for all $i$ and $j$.

**Assumption 5.** In the model Eq. (8), the following conditions hold:

1. For all $i \in [M]$, the function $h_i$ satisfies $\|\mathbb{E}_{U \sim N(0, I_k)}[h_i(U)U]\| = O(1)$;

2. For all $u, v \in \mathbb{R}^k$ and $i \in [M]$, it holds that $|h_i(u) - h_i(v)| = O(\|u - v\|)$;

3. Errors $y_{ij} \mid x_{ij} - \mathbb{E}[y_{ij} \mid x_{ij}]$ are independent sub-gaussian with constant variance proxy.

---

[1]A random variable $\xi \in \mathbb{R}$ is sub-Gaussian with variance proxy $\sigma^2$, denoted by $\xi \sim \text{subG}(\sigma^2)$, if $\mathbb{E}[\exp(t(\xi - \mathbb{E}\xi))] \leq \exp(\sigma^2 t^2/2)$ for any $t \in \mathbb{R}$. A random vector $\xi \in \mathbb{R}^d$ is sub-Gaussian with variance proxy $\sigma^2$, denoted by $\xi \sim \text{subG}_d(\sigma^2)$, if $u^\mathsf{T}\xi \sim \text{subG}(\sigma^2)$ for any $u \in \mathbb{S}^{d-1}$.

| Reference | Method | General cases | Well-represented |
|---|---|---|---|
| *Upper Bound:* | | | |
| (Du et al., 2021) | ERM | - | $O\left(\sqrt{\frac{dk^2}{N}}\right)$ |
| (Tripuraneni et al., 2021) | MoM | $O\left(\sqrt{\frac{d}{N\lambda_k^2}}\right)$ | $O\left(\sqrt{\frac{dk^2}{N}}\right)$ |
| (Duchi et al., 2022) | Spectral | $O\left(\sqrt{\frac{d}{N\lambda_k^2}}\right)$ | $O\left(\sqrt{\frac{dk^2}{N}}\right)$ |
| (Collins et al., 2021) | AltMin | - | $O\left(\sqrt{\frac{dk^2}{N}}\right)$ |
| (Thekumparampil et al., 2021) | AltMin | $O\left(\sqrt{\frac{dk\lambda_1}{N\lambda_k^2}}\right)$ | $O\left(\sqrt{\frac{dk^2}{N}}\right)$ |
| (Zhang et al., 2024) | AltMin | - | $O\left(\sqrt{\frac{dk}{N}}\right)$ |
| (Chua et al., 2021) | AdaptRep | - | $O\left(\sqrt{\frac{dk}{N}}\right)$ |
| (Duan & Wang, 2023) | ARMUL | - | $O\left(\sqrt{\frac{dk^2}{N}}\right)$ |
| (Tian et al., 2025) | Spectral | - | $O\left(\sqrt{\frac{dk}{N}}\right)$ |
| (Niu et al., 2024) | Spectral | $O\left(\sqrt{\frac{d\lambda_1}{N\lambda_k^2}} + \sqrt{\frac{Md}{N^2\lambda_k^2}}\right)$ | $\Theta\left(\sqrt{\frac{dk}{N}} + \sqrt{\frac{Mdk^2}{N^2}}\right)$ |
| *Lower Bound:* | | | |
| (Tripuraneni et al., 2021) | - | $\Omega\left(\sqrt{\frac{1}{N\lambda_k}} + \sqrt{\frac{dk}{N}}\right)$ | $\Omega\left(\sqrt{\frac{dk}{N}}\right)$ |
| (Niu et al., 2024) | - | $\Omega\left(\sqrt{\frac{d}{N\lambda_k}} + \sqrt{\frac{Md}{N^2\lambda_k^2}}\right)$ | |

*Table 3.* A summary of the high-probability statistical rate for estimating $B^\star$. Here "general cases" refer to cases with a general source diversity matrix $D$, while "well-represented" cases assume $\lambda_1 = \Theta(\lambda_k) = \Theta(1/k)$. The abbreviations for the methods are as follows: "ERM": empirical risk minimization; "MoM": method-of-moments estimator; "AltMin": alternating minimization algorithm initialized from MoM; "AdaptRep": adaptive representation learning; "ARMUL": adaptive and robust multi-task learning.

## D. Extended Experiments

### D.1. Synthetic Data Setup

For our synthetic distributed data experiments, we generate $n_i$ samples $(x_{ij}, y_{ij})$ for each source $i \in [M]$ according to $y_{ij} = x_{ij}^\top B^\star \alpha_i^\star + \epsilon_{ij}$. Here, $B^\star \in \mathbb{R}^{d \times k}$ represents the shared subspace and $\alpha_i^\star \in \mathbb{R}^k$ are source-specific coefficients. By default, we set $M = 100$, $d = 30$, and $k = 6$.

The shared basis $B^\star$ is sampled from a Haar distribution, with features $x_{ij} \sim \mathcal{N}(0, \frac{1}{d}I)$ and noise $\epsilon_{ij} \sim \mathcal{N}(0, 1)$. To model a heterogeneous setup, source sample sizes $n_i$ are drawn i.i.d. from $[\frac{d}{3}, d]$. This regime ensures $n_i < d$, rendering purely local learning insufficient and necessitating robust data screening.

We consider two distinct regimes for the coefficients $\alpha_i$:

- **Clustered Coefficients:** Each source is assigned to a group $G_i \in \{1, 2\}$ with $P(G_i = 1) = g$ (default $g = 0.2$). If $G_i = 1$, then $\alpha_i \sim \mathcal{N}(0, \text{diag}(\mathbf{1}_{k/2}, \mathbf{0}_{k/2}))$; otherwise, $\alpha_i \sim \mathcal{N}(0, \text{diag}(\mathbf{0}_{k/2}, \mathbf{1}_{k/2}))$.

- **Heterogeneous Gaussian:** Coefficients are drawn as $\alpha_i \sim \mathcal{N}(0, \Psi_i)$, where $\Psi_i$ is a random PSD matrix. We generate each $\Psi_i$ by first sampling a random matrix $A_i$ with i.i.d. entries uniformly in $[0, 1]$, forming $\Psi_i = (A_i + A_i^T)/2 + 3I_k$, and normalizing its trace so that the average eigenvalue is one.

### D.2. Real-World Dataset Setup

To assess practical applicability, we evaluate our method on the ACSIncome (Ding et al., 2021) and CelebA (Liu et al., 2015) datasets. These benchmarks provide natural data partitions that reflect real-world distribution shifts and allow us to test our method across regimes of varying complexity. For all tasks, we adopt FedRep (Collins et al., 2021) as the base estimator and report the final classification accuracy averaged over the entire source population. We run our algorithm with varying assumed values of dimension $k$.

### D.3. ACSIncome

The ACSIncome dataset is used to predict whether an individual's annual income exceeds \$50,000. The dataset contains approximately 1.6 million samples with low-dimensional tabular features ($d = 10$). We partition the data geographically by state, where each state represents a source, to simulate natural heterogeneity.

Formally, for source $s$, let $x_i \in \mathbb{R}^{10}$ denote the raw tabular features of the $i$-th individual, $y_i \in \{0, 1\}$ be the income label, and $\theta_s$ be the source-specific logistic regression parameter. This task is modeled using federated logistic regression.

### D.4. CelebA

For high-dimensional data, we utilize the CelebA dataset for smile classification. Following the LEAF benchmark (Caldas et al., 2018), we select a 10% subset of the population, resulting in approximately 20,000 samples partitioned by individual identity, where each identity represents a source. The inputs are $224 \times 224$ RGB images.

Given the high dimensionality, we employ a Vision Transformer (ViT-Tiny) (Dosovitskiy, 2020) pre-trained on ImageNet to extract a lower-dimension signal ($d = 192$) from the images prior to subset selection. Specifically, we extract the activations from the penultimate layer corresponding to the 'cls' token (Dosovitskiy, 2020).

Formally, for source $s$, $x_i \in \mathbb{R}^{192}$ is the ViT-Tiny embedding of the $i$-th image, $y_i \in \{0, 1\}$ is the smile label, and $\theta_s$ is the source-specific classification head. We then fine-tune the model with the selected sources.

### D.5. Nonlinear Feature Extractors

**Ablation on ACSIncome.** While our theoretical guarantees are developed for linear models, the source-screening procedure is agnostic to the downstream learner and can be applied as a preprocessing step prior to any training regime. To validate this, we conduct additional experiments on ACSIncome, replacing the logistic regression model with a two-layer neural network with ReLU activations as a nonlinear feature extractor and varying the hidden layer dimension ($d_h \in \{64, 256\}$). Source selection is performed using our screening algorithm before initiating `FedRep` training of the neural network. All other experimental details follow Section 6.2.

| | Mean Accuracy (%) | |
|---|---|---|
| Selection Method | $d_h = 64$ | $d_h = 256$ |
| Full Population | 79.91 | 79.49 |
| Random Selection | 78.08 | 77.87 |
| Power-of-Choice | 79.70 | 79.41 |
| Ours | **80.13** | **79.96** |

*Table 4.* Classification accuracy on ACSIncome with a two-layer neural network feature extractor at hidden dimensions $d_h \in \{64, 256\}$. Our source-screening procedure consistently outperforms all baselines, demonstrating that the benefits extend beyond the linear regime.

As shown in Table 4, our screening procedure yields consistent accuracy gains over full client training and other subset selection baselines, even when training a nonlinear model. This suggests the benefits of source screening naturally transfer to more complex model classes.

### D.6. Reweighting Clients Instead of Removal

A natural alternative to client subsampling is to reweight client contributions during distributed averaging rather than removing them entirely. We address this question both theoretically and empirically.

**Theoretical perspective.** Existing upper bounds do not readily extend to sample reweighting. To see this, consider the special class of problem instances in Section 4.1. Appropriate reweighting will easily make the reweighted diversity matrix $D$ well-conditioned. However, reweighting also directly impacts the behavior of subspace estimators. In particular, existing analysis of those estimators relies on the statistical concentration of all used samples as a whole, a structure that sample reweighting may undermine.

For the general collection of clients, departing from the results in Section 4.2.2, appropriate sample reweighting may not exist. The analysis in Section 4.2.2 heavily relies on (i) column selection (Algorithm 1, lines 12–20) and (ii) the concatenation of

the selected columns (Algorithm 1, line 29). Sample reweighting significantly affects the correctness of both steps. Roughly speaking, assuming matrix $A$ is standardized (i.e., all columns have $\ell_2$ norm equal to 1), Theorem 4 guarantees that, at each iteration, the outer product of the selected columns, taken as a whole, has a constant-level condition number. Sample reweighting undermines this column standardization structure, affecting the construction of the matrix $H$ (Appendix I). In particular, instead of the standard subtraction, one may need to subtract a diagonal matrix with entries closely related to the weights. Following the roadmap of existing analysis, the condition number of the weighted version of the relevant matrix may no longer remain constant. Moreover, unless a more refined version of Weyl's inequalities is available, the accumulation of weights (analogous to matrix concatenation in the original analysis) may fail to preserve a constant condition number.

**Empirical comparison.** Numerically, sample reweighting may help reduce noise compared with pure removal. We train a two-layer neural network on the ACS Income Dataset and evaluate reweighting as an alternative to client subsampling. Specifically, we compute the Grothendieck Factorization of the diversity matrix (i.e., line 19 of our empirical subpopulation search, applied to all clients) and use the inverse of the client $d_j$ values to weight their contribution during distributed averaging with FedRep. We consider two paradigms: **V1**, which only considers $d_j$ with weights $\frac{d_j^{-1}}{\sum_{j'} d_{j'}^{-1}}$, and **V2**, which also incorporates client data volume $m_j$ (as in standard federated averaging) with weights $\frac{m_j d_j^{-1}}{\sum_{j'} m_{j'} d_{j'}^{-1}}$. As shown in Table 5, while reweighting strategies may outperform full population training with lower variance across seeds, they achieve lower average accuracy than source subsampling.

*Table 5.* Comparison of client selection strategies on the ACS Income Dataset (two-layer neural network).

| Selection Method | Mean Accuracy | Std. Accuracy |
|---|---|---|
| Subsampling | 79.96 | 0.21 |
| Reweight V1 | 79.38 | 0.03 |
| Reweight V2 | 79.58 | 0.06 |
| Full Population | 79.49 | 0.06 |

# E. Empirical Algorithm

---

**Algorithm 2** Empirical Subpopulation Search

---
1: **Input:** Matrices of split local averaging $\bar{Z} = [\bar{z}_1, \cdots, \bar{z}_M]$ and $\widetilde{Z} = [\widetilde{z}_1, \cdots, \widetilde{z}_M]$ such that $\|\bar{Z}\|^2 \lesssim \frac{M}{k}$ and $\|\widetilde{Z}\|^2 \lesssim \frac{M}{k}$.
   An absolute constant $c^*$ for which $320(c^* + \sqrt{2c^*}) \leq 0.5$. A target success rate $\delta \in (0, 1)$;
2: **Output:** A set of column indices $\mathcal{S} \subseteq [M]$.

3: $\widehat{A} \leftarrow \bar{Z} + \widetilde{Z}$.
4: Compute st.rank($\widehat{A}$).
5: **if** $c^* \cdot$ st.rank($\widehat{A}$) $< 1$ **then**
6:     **Return** $\emptyset$, and display *"low stable rank"*;
7: **end if**
8: Let $s = \lceil c^* \cdot \text{st.rank}(\bar{Z} + \widetilde{Z}) \rceil$;
9: Compute $\lambda_{\min}(\widehat{A}\widehat{A}^\top)$;
10: $A_1 \leftarrow \widehat{A}$, and $\mathcal{S} \leftarrow \emptyset$;
11: Compute $\|A_1\|^2$;
12: $\widetilde{c} \leftarrow \frac{\|A_1\|^2}{M/k}$;
13: **for** $t = 1, \cdots, \lceil \lambda_{\min}(\widehat{A}\widehat{A}^\top) \rceil$ **do**
14:     **if** st.rank($A_t$) $\geq \frac{k}{2\widetilde{c}}$ **then**
15:         **for** $\ell = 1, ..., \log_{8/7} \frac{\lambda_{\min}(\widehat{A}\widehat{A}^\top)}{\delta}$ **do**
16:             Draw a uniformly random set $\widetilde{\mathcal{S}}_t$ with cardinality $s$;
17:             Compute the $H_{\widetilde{\mathcal{S}}_t \times \widetilde{\mathcal{S}}_t} = A_{\widetilde{\mathcal{S}}_t}^\top A_{\widetilde{\mathcal{S}}_t} - I_s$;
18:             **if** $\|H_{\widetilde{\mathcal{S}}_t \times \widetilde{\mathcal{S}}_t}\|_{\infty \to 1} \leq \frac{s}{8}$ **then**
19:                 Perform Grothendieck Factorization on $H_{\widetilde{\mathcal{S}}_t \times \widetilde{\mathcal{S}}_t}$ to obtain $H_{\widetilde{\mathcal{S}}_t \times \widetilde{\mathcal{S}}_t} = D_t T_t D_t$;
20:                 Let $\mathcal{S}_t = \{j : d_{jt}^2 \leq 2/s, j \in \widetilde{\mathcal{S}}_t\}$, where $d_{jt}$ is the $j$-th diagonal entry of $D_t$;
21:                 **Break**.    {break the inner for-loop and proceed to execute the remaining commands}
22:             **end if**
23:         **end for**
24:         Remove columns in $\mathcal{S}_t$ from the matrix $A_t$ to obtain $A_{t+1}$;
25:     **else**
26:         **Break** {break the outer for-loop and proceed to execute the remaining commands}
27:     **end if**
28: **end for**
29: Set $t^* \leftarrow t$; {$t^*$ reads out the interation upon which the for-loop terminates.}
30: **if** st.rank($A_{t^*}$) $\geq \frac{k}{2\widetilde{c}}$ **then**
31:     $\mathcal{S} \leftarrow \cup_{r=1}^{t^*} \mathcal{S}_r$;
32: **else**
33:     $\mathcal{S} \leftarrow \cup_{r=1}^{t^*-1} \mathcal{S}_r$;
34: **end if**
35: **Return** $\mathcal{S}$.

---

# F. Auxiliary Lemmas and Theorems in Section 4.2

**Theorem 6** (Weyl's inequalities). *(Bhatia, 2013, Theorem III.2.1) Let $P$ and $Q$ by $d \times d$ real symmetric matrices. Let eigenvalues in nonincreasing order: $\lambda_1(P) \geq \cdots \geq \lambda_d(P)$. Similarly, define eigenvalues for $Q$ and $P + Q$. For all $i, j$ such that $i + j - 1 \leq d$:*

$$\lambda_{i+j-1}(P + Q) \leq \lambda_i(P) + \lambda_j(Q).$$

*For all $i, j$ such that $i + j - 1 \geq d$:*

$$\lambda_{i+j-d}(P + Q) \geq \lambda_i(P) + \lambda_j(Q).$$

## G. Proof of Theorem 2

*Proof.* Since $n_i = n$ for all $i \in [M]$, we have $N = Mn$, that is $n = \frac{N}{M}$. From Table 1, we know that when $\lambda_k \geq \frac{1}{n}$, the minimax lower bound in estimating $B^\star$ is

$$\sqrt{\frac{d}{N\lambda_k}} + \sqrt{\frac{Md}{N^2\lambda_k^2}} \asymp \sqrt{\frac{d}{N\lambda_k}},$$

recalling that $\lambda_k = \lambda_{\min}(D)$. From the analysis in (Niu et al., 2024), we know that when training over the entire population, the error upper bound is

$$\sqrt{\frac{d\lambda_1}{N\lambda_k^2}} + \sqrt{\frac{Md}{N^2\lambda_k^2}} \asymp \sqrt{\frac{d\lambda_1}{N\lambda_k^2}}.$$

Let's consider collaboratively estimating the subspace using the split local averaging algorithm in (Niu et al., 2024). Let $N' = n|\mathcal{S}|$, $D' = \frac{1}{|\mathcal{S}|}\sum_{i\in\mathcal{S}} \alpha_i^\star(\alpha_i^\star)^\top$. Let $\lambda_1' = \lambda_{\max}(D')$ and $\lambda_k' = \lambda_{\min}(D')$. Since we know $\mathcal{S}$ satisfies Definition 2, then $\lambda_1' \asymp \lambda_k' \asymp \frac{1}{k}$, and

$$N' = n|\mathcal{S}| \asymp nk\lambda_{\min}(AA^\top) = nkM \cdot \lambda_{\min}\left(\frac{1}{M}AA^\top\right)$$

$$= nkM \cdot \lambda_{\min}(D) = nkM\lambda_k.$$

Thus, we have

$$\sqrt{\frac{d\lambda_1'}{N'(\lambda_k')^2}} \asymp \sqrt{\frac{d}{N'\lambda_k'}} \asymp \sqrt{\frac{d}{nkM\lambda_k\frac{1}{k}}} = \sqrt{\frac{d}{N\lambda_k}}, \tag{9}$$

matching the lower bound. Hence, the achieved rate is minimax optimal.

$\square$

## H. Proof of Theorem 3

*Proof.* We first show that $\lambda_{\min}(AA^\top)$ cannot exceed $M/k$.

$$\mathrm{Tr}(AA^\top) = \mathrm{Tr}(\sum_{i=1}^{M} \alpha_i^\star(\alpha_i^\star)^\top) = M = \sum_{i=1}^{k} \lambda_i(AA^T).$$

Then, we have $\lambda_{\min}(AA^\top) \leq \frac{M}{k}$.

Then, we consider two cases: (1) $\lambda_{\min}(AA^\top) = \Theta(M/k)$, and (2) $\lambda_{\min}(AA^\top) = o(M/k)$.

In the first case, the original matrix $A$ is well-conditioned, and we can choose $\mathcal{S} = [M]$. Specifically,

$$\kappa(AA^\top) = \frac{\|AA^\top\|}{\lambda_{\min}(AA^\top)} = \frac{CM/k}{cM/k} = \Theta(1),$$

where $C > 0, c > 0$ are two absolute constants. Throughout this paper, the same notation $C$ and $c$ may mean different specific values, yet they do not scale with key parameters such as $M$, $k$, and $\lambda_{\min}(AA^\top)$. In addition, $|\mathcal{S}| = M = \Theta(k \cdot \lambda_{\min}(AA^\top))$.

Henceforth, we focus on proving case 2, i.e., $\lambda_{\min}(AA^\top) = o(M/k)$. We construct a conceptual algorithm (Algorithm 3) to establish the existence of the desired subpopulation. Since our goal is purely existential, neither the computational complexity nor the practical implementability of Algorithm 3 plays any role in the proof.

---

**Algorithm 3** Existence of a Good Subpopulation

---

1: **Input:** A full-row-rank standardized matrix $A = [\alpha_1^*, \alpha_2^*, \cdots, \alpha_M^*]$, where $\|\alpha_i^*\|_2 = 1$ for $i = 1, \cdots, M$, and $\|A\|^2 \lesssim \frac{M}{k}$

2: **Output:** A set of column indices $\mathcal{S} \subseteq [M]$.

3: $A_1 \leftarrow A$, and $\mathcal{S} \leftarrow \emptyset$;

4: Compute $\|A_1\|^2$;

5: $\widetilde{c} \leftarrow \frac{\|A_1\|^2}{M/k}$;

6: **for** $t = 1, \cdots, \lceil \lambda_{\min}(AA^\top) \rceil$ **do**

7:    **if** st.rank$(A_t) \geq \frac{k}{2\widetilde{c}}$ **then**

8:       Find $\mathcal{S}_t$, the subset of columns of matrix $A_t$ promised by Theorem 4;

9:       Remove columns whose indices are in the subset $\mathcal{S}_t$ from the matrix $A_t$ to obtain $A_{t+1}$;

10:    **else**

11:       **Break** {break out the for-loop and proceed to execute the remaining commands}

12:    **end if**

13: **end for**

14: Set $t^* \leftarrow t$; {$t^*$ reads out the iteration upon which the for-loop terminates.}

15: **if** st.rank$(A_{t^*}) \geq \frac{k}{2\widetilde{c}}$ **then**

16:    $\mathcal{S} \leftarrow \cup_{r=1}^{t^*} \mathcal{S}_r$;

17: **else**

18:    $\mathcal{S} \leftarrow \cup_{r=1}^{t^*-1} \mathcal{S}_r$;

19: **end if**

20: **Return** $\mathcal{S}$.

---

Note that the if-clause in the for-loop is executed at least once because st.rank$(A_1) = \frac{k}{\widetilde{c}}$ as per the definition of $\widetilde{c}$. Hence, $\mathcal{S} \neq \emptyset$.

For any set $\mathcal{S}' \subseteq [M]$, let $A_{\mathcal{S}'}$ denote the submatrix of $A$ with columns restricted to the set $\mathcal{S}'$. For any non-overlapping subsets $\mathcal{S}'$ and $\mathcal{S}''$, from the Weyl's inequalities (Theorem 6), we know that

$$\lambda_1 \left( A_{\mathcal{S}'} A_{\mathcal{S}'}^\top + A_{\mathcal{S}''} A_{\mathcal{S}''}^\top \right) \leq \lambda_1 \left( A_{\mathcal{S}'} A_{\mathcal{S}'}^\top \right) + \lambda_1 \left( A_{\mathcal{S}''} A_{\mathcal{S}''}^\top \right),$$

$$\lambda_k \left( A_{\mathcal{S}'} A_{\mathcal{S}'}^\top + A_{\mathcal{S}''} A_{\mathcal{S}''}^\top \right) \geq \lambda_k \left( A_{\mathcal{S}'} A_{\mathcal{S}'}^\top \right) + \lambda_k \left( A_{\mathcal{S}''} A_{\mathcal{S}''}^\top \right).$$

When $\lambda_k \left( A_{\mathcal{S}'} A_{\mathcal{S}'}^\top \right) > 0$ and $\lambda_k \left( A_{\mathcal{S}''} A_{\mathcal{S}''}^\top \right) > 0$, we have

$$
\begin{aligned}
\kappa \left( A_{\mathcal{S}'} A_{\mathcal{S}'}^\top + A_{\mathcal{S}''} A_{\mathcal{S}''}^\top \right) &= \frac{\lambda_1 \left( A_{\mathcal{S}'} A_{\mathcal{S}'}^\top + A_{\mathcal{S}''} A_{\mathcal{S}''}^\top \right)}{\lambda_k \left( A_{\mathcal{S}'} A_{\mathcal{S}'}^\top + A_{\mathcal{S}''} A_{\mathcal{S}''}^\top \right)} \\
&\leq \frac{\lambda_1 \left( A_{\mathcal{S}'} A_{\mathcal{S}'}^\top \right) + \lambda_1 \left( A_{\mathcal{S}''} A_{\mathcal{S}''}^\top \right)}{\lambda_k \left( A_{\mathcal{S}'} A_{\mathcal{S}'}^\top \right) + \lambda_k \left( A_{\mathcal{S}''} A_{\mathcal{S}''}^\top \right)} \\
&= \frac{\kappa_{\mathcal{S}'} \lambda_k \left( A_{\mathcal{S}'} A_{\mathcal{S}'}^\top \right) + \kappa_{\mathcal{S}''} \lambda_k \left( A_{\mathcal{S}''} A_{\mathcal{S}''}^\top \right)}{\lambda_k \left( A_{\mathcal{S}'} A_{\mathcal{S}'}^\top \right) + \lambda_k \left( A_{\mathcal{S}''} A_{\mathcal{S}''}^\top \right)} \\
&\leq \max\{\kappa_{\mathcal{S}'}, \kappa_{\mathcal{S}''}\},
\end{aligned}
\tag{10}
$$

where $\kappa_{\mathcal{S}'}$ and $\kappa_{\mathcal{S}''}$ are the condition numbers of matrices $A_{\mathcal{S}'} A_{\mathcal{S}'}^\top$ and $A_{\mathcal{S}''} A_{\mathcal{S}''}^\top$, respectively.

Without loss of generality, assume that $\mathcal{S} = \cup_{r=1}^{t^*} \mathcal{S}_r$; the other case can be shown analogously. We have

$$1 \leq \kappa \left( \sum_{i \in \mathcal{S}} \alpha_i^\star (\alpha_i^\star)^\top \right) = \kappa \left( \sum_{i \in \cup_{r=1}^{t^*} \mathcal{S}_r} A_{\mathcal{S}_r} (A_{\mathcal{S}_r})^\top \right) \overset{(a)}{\leq} \max_{r \in \{1, \ldots, t^*\}} \kappa \left( A_{\mathcal{S}_r} (A_{\mathcal{S}_r})^\top \right) \leq 3,$$

where inequality (a) holds by repeatedly applying the arguments in Eq. (10), and the last inequality follows from Theorem 4.

To complete the proof of case 2, it remains to show $|\mathcal{S}| = \Omega \left( k \cdot \lambda_{\min}(AA^\top) \right)$. When $\lambda_{\min}(AA^\top) = O(1)$, we have

$$|\mathcal{S}| \geq |\mathcal{S}_1| \overset{(a)}{=} \Theta(k) = \Omega(k \cdot \lambda_{\min}(AA^\top)),$$

where equality (a) is ensured by Theorem 4. When $\lambda_{\min}(AA^\top) = \omega(1)$, we need to show that $t^* = \Theta(\lambda_{\min}(AA^\top))$. To see this, we know from Theorem 4 that at each $t \leq t^*$, we obtain $\mathcal{S}_t$ of size $\Theta(k)$ independently of $M$. Then, $|\mathcal{S}| = t^*\Theta(k) = O(k\lambda_{\min}(AA^\top)) = o(M)$ regardless of the choice of $t^*$, which implies $|\mathcal{S}| \leq \frac{1}{2}M$ when $M$ is sufficiently large. Since $A_t A_t^\top$ is positive semi-definite, it holds that $\|A_t\|^2 \leq \|A\|^2$ for each $t = 1, \cdots, t^*$. Hence, for each $t = 1, \cdots, \lceil \lambda_{\min}(AA^\top) \rceil$, we have

$$\frac{\|A_t\|_F^2}{\|A_t\|^2} \geq \frac{\|A_t\|_F^2}{\|A\|^2} = \frac{M - |\cup_{r=1}^{t-1} \mathcal{S}_r|}{\|A\|^2} \geq \frac{M - |\mathcal{S}|}{\|A\|^2} \geq \frac{M/2}{\widetilde{c}M/k} = \frac{k}{2\widetilde{c}}.$$

By the termination criterion of the for-loop in Algorithm 3, we know it will not terminate before round $\lceil \lambda_{\min}(AA^\top) \rceil$. Thus, $t^* = \lceil \lambda_{\min}(AA^\top) \rceil$, resulting in

$$|\mathcal{S}| = \Theta\left(k\lambda_{\min}(AA^\top)\right).$$

$\square$

# I. Proof of Theorem 4

Let $\delta \in [0, 1]$. Let $P_\delta$ be a random $M \times M$ diagonal matrix where exactly $s = \lfloor \delta M \rfloor$ entries equal one and the rest equal zero. With a little abuse of notation, we treat $AP_\delta$ as a random $s$-column submatrix of $A$ by ignoring the zeroed columns. Let $p, q \in [1, +\infty]$. The matrix norm $\| \cdot \|_{p \to q}$ is defined as

$$\|A\|_{p \to q} = \max_{x \in \mathbb{R}^M : \|x\|_p = 1} \|Ax\|_q.$$

Let $R_\delta$ be a random $M \times M$ diagonal matrix whose diagonal entries are independent 0-1 Bernoulli random variables with common mean $\delta$.

**Proposition 1** ((Tropp, 2009)). *For any $p, q \in [1, +\infty]$, and for any matrix $A$ with $M$ columns, it holds that*

$$\mathbb{E}\|AP_\delta\|_{p \to q} \leq 2\mathbb{E}\|AR_\delta\|_{p \to q}.$$

*For each $M \times M$ matrix $H$, it holds that*

$$\mathbb{E}[\|P_\delta H P_\delta\|_{p \to q}] \leq 2\mathbb{E}\|R_\delta H R_\delta\|_{p \to q}.$$

Let

$$\|H\|_{\mathrm{col}} = \sum_{j=1}^M \|He_j\|_2$$

be the *column norm* of $H$, where $\{e_j\}_{j=1}^M$ is the collection of standard basis of $\mathbb{R}^M$.

**Lemma I.1** ((Rudelson & Vershynin, 2007)). *Fix $\delta \in [0, 1]$. Suppose that $H \in \mathbb{R}^{M \times M}$. Then*

$$\mathbb{E}\|R_\delta H R_\delta\|_{\infty \to 1} \leq 20 \left[\delta^2 \|H - diag(H)\|_{\infty \to 1} + \delta^{3/2} \left(\|H\|_{col} + \|H^*\|_{col}\right) + \delta \|diag(H)\|_{\infty \to 1}\right].$$

**Lemma I.2.** *(Pisier et al., 1986) Each matrix $G$ can be factorized as $G = D_1 T D_2$ such that*

- *$D_i$ is a non-negative, diagonal matrix with $trace(D_i^2) = 1$ for $i = 1, 2$, and*

- *$\|G\|_{\infty \to 1} \leq \|T\| \leq 2\|G\|_{\infty \to 1}$.*

*When $G$ is Hermitian, we can take $D_1 = D_2$.*

## I.1. Proof of Theorem 4

With the above notation and auxiliary results in place, we are ready to prove Theorem 4. The proof presented here largely follows the analysis of (Tropp, 2009), but with all multiplicative constants made explicit. This explicit treatment is necessary to highlight a minor caveat that was previously overlooked and to underscore the need to impose the condition $\text{st.rank}(A) = \omega(1)$ with respect to $k$ and $M$. The original statement and proof in (Bourgain & Tzafriri, 1987) may not require this condition; however, their argument is rooted in a functional-analytic perspective and does not provide algorithmic insight.

Let $H = A^\top A - I_M$, where $I_M \in \mathbb{R}^{M \times M}$ is the identity matrix. Since each column of $A$ has $\ell_2$ norm 1, the diagonal entries of $A^\top A$ are all ones. Thus, $\text{diag}(H) = \mathbf{0}$. Applying Lemma I.1, we know that for any $\delta \in (0, 1)$,

$$\mathbb{E}\|R_\delta H R_\delta\|_{\infty \to 1} \leq 20 \left[ \delta^2 \|H\|_{\infty \to 1} + 2\delta^{3/2} \|H\|_{\text{col}} \right].$$

Recall that by definition, $\|H\|_{\infty \to 1} = \max_{x \in \mathbb{R}^M : \|x\|_\infty = 1} \|Hx\|_1$. Furthermore, we have, for any $x \in \mathbb{R}^M$,

$$\|Hx\|_1 \leq \sqrt{M}\|Hx\| \leq \sqrt{M}\|H\|\|x\| \leq \sqrt{M}\|H\|(\sqrt{M}\|x\|_\infty) = M\|H\|\|x\|_\infty.$$

Then,

$$\|H\|_{\infty \to 1} = \max_{x \in \mathbb{R}^M : \|x\|_\infty = 1} \|Hx\|_1 \leq \max_{x \in \mathbb{R}^M : \|x\|_\infty = 1} M\|H\|\|x\|_\infty = M\|H\|. \tag{11}$$

Therefore, by Eq. (11),

$$\|H\|_{\infty \to 1} \leq M\|H\| \leq M \max\left\{\|A\|^2 - 1, 1\right\} \leq M\|A\|^2. \tag{12}$$

Meanwhile, we have

$$\|H\|_{\text{col}} \overset{(a)}{<} \|A^\top A\|_{\text{col}} = \sum_{j=1}^M \|A^\top a_j\|_2 \leq M\|A\|, \tag{13}$$

where $a_j$ is the $j$-th column of $A$. (a) holds because removing the diagonal entries will only decrease the column norm. The last inequality holds because $A$ is standardized, meaning that $\|a_j\| = 1$. Then, by Eq. (12) and Eq. (13), we have

$$\mathbb{E}\|R_\delta H R_\delta\|_{\infty \to 1} \leq 20 \left[ \delta^2 \|H\|_{\infty \to 1} + 2\delta^{3/2} \|H\|_{\text{col}} \right]$$
$$\leq 20 \left[ \delta^2 M\|A\|^2 + 2\delta^{3/2} M\|A\| \right]. \tag{14}$$

Let $\delta M = |\mathcal{S}| = \lceil c \cdot \text{st.rank}(A) \rceil$. Then, the upper bound Eq. (14) can be written as,

$$\mathbb{E}\|R_\delta H R_\delta\|_{\infty \to 1} \leq 20|\mathcal{S}| \left[ \delta\|A\|^2 + 2\delta^{1/2}\|A\| \right].$$

It turns out that being able to control the absolute constant $c$ in the upper bound of $\mathbb{E}\|R_\delta H R_\delta\|_{\infty \to 1}$ is crucial in guaranteeing that the condition number of the obtained subset is $\Theta(1)$. If $\text{st.rank}(A) = \Theta(1)$, $|\mathcal{S}| = \lceil c \cdot \text{st.rank}(A) \rceil = \Theta(c)$, however, we then require $c$ being small such that Eq. (17) holds. Therefore, we are not always able to control the absolute constant by adjusting $c$ without violating the requirement that $|\mathcal{S}| \geq 1$. This explains why requiring $\text{st.rank}(A) = \omega(1)$ w.r.t. $k$ and $M$.

When $k$ and $M$ are sufficiently large so that $\text{st.rank}(A)$ is sufficiently large, we have

$$c \cdot \text{st.rank}(A) \leq \delta M = |\mathcal{S}| \leq c \cdot \text{st.rank}(A) + 1 \leq 2c \cdot \text{st.rank}(A)$$
$$= \frac{2c\|A\|_F^2}{\|A\|^2} = \frac{2cM}{\|A\|^2},$$

i.e., $\delta \leq \frac{2c}{\|A\|^2}$. So,

$$\mathbb{E}\|R_\delta H R_\delta\|_{\infty \to 1} \leq 20\delta M \left( \delta\|A\|^2 + 2\delta^{1/2}\|A\| \right) \leq 20\delta M \left( 2c + 2\sqrt{2c} \right). \tag{15}$$

By Proposition 1, we have

$$\mathbb{E}\|P_\delta H P_\delta\|_{\infty\to 1} \leq 2\mathbb{E}\|R_\delta H R_\delta\|_{\infty\to 1} \leq 40\delta M\left(2c + 2\sqrt{2c}\right). \tag{16}$$

Then, there must exist one realization of $P_\delta$, i.e., one subset $\mathcal{S}_0$, such that

$$\left\|A_{\mathcal{S}_0}^\top A_{\mathcal{S}_0} - I_{|\mathcal{S}_0|}\right\|_{\infty\to 1} \leq 40\delta M\left(2c + 2\sqrt{2c}\right).$$

By Lemma I.2, applying Grothendieck factorization to $\left(A_{\mathcal{S}_0}^\top A_{\mathcal{S}_0} - I_{|\mathcal{S}_0|}\right)$, a Hermitian matrix, we have

$$A_{\mathcal{S}_0}^\top A_{\mathcal{S}_0} - I_{|\mathcal{S}_0|} = DTD,$$

where $D$ is a non-negative, diagonal matrix with $\operatorname{trace}(D^2) = 1$, and $\|T\| \leq 2\|A_{\mathcal{S}_0}^\top A_{\mathcal{S}_0} - I_{|\mathcal{S}_0|}\|_{\infty\to 1}$. Define $\mathcal{S} \subseteq \mathcal{S}_0$ as

$$\mathcal{S} = \left\{i : d_{ii}^2 \leq \frac{2}{|\mathcal{S}_0|}, i \in \mathcal{S}_0\right\}.$$

Note that it must be true that $|\mathcal{S}| \geq \frac{|\mathcal{S}_0|}{2}$. Otherwise, $1 \geq \sum_{i\in\mathcal{S}_0\setminus\mathcal{S}} d_{ii}^2 > \frac{2}{|\mathcal{S}_0|}\frac{|\mathcal{S}_0|}{2} = 1$, a contradiction. Let $\widehat{D}_\mathcal{S} \in \mathbb{R}^{|\mathcal{S}_0|\times|\mathcal{S}_0|}$ be a diagonal matrix that retains the original diagonal entries indexed by $\mathcal{S}$ and sets the remaining diagonal entries to zero. It is not hard to see $\|\widehat{D}_\mathcal{S}\|^2 \leq \frac{2}{|\mathcal{S}_0|}$. Though $A_\mathcal{S}^\top A_\mathcal{S} - I_{|\mathcal{S}|} \neq \widehat{D}_\mathcal{S}T\widehat{D}_\mathcal{S}$, they are equal in terms of spectral norm:

$$\begin{aligned}
\|A_\mathcal{S}^\top A_\mathcal{S} - I_{|\mathcal{S}|}\| &= \|\widehat{D}_\mathcal{S}T\widehat{D}_\mathcal{S}\| \\
&\leq \|T\|\|\widehat{D}_\mathcal{S}\|^2 \\
&\leq \frac{2}{|\mathcal{S}_0|}2\|A_{\mathcal{S}_0}^\top A_{\mathcal{S}_0} - I_{|\mathcal{S}_0|}\|_{\infty\to 1} \\
&\leq \frac{2}{|\mathcal{S}_0|}2\cdot 40|\mathcal{S}_0|(2c + 2\sqrt{2c}) \\
&= 320\left(c + \sqrt{2c}\right).
\end{aligned}$$

Since $c$ is determined by the choice of the target size $|\mathcal{S}|$, we can partially control its value to make the above upper bound small. However, a valid choice of $c$ must ensure that the resulting target size satisfies $\delta M \geq 1$ and that $c \cdot \operatorname{st.rank}(A) \geq 1$, which in turn requires $k$ and $M$ to be sufficiently large. In particular, we will choose $c$ so that

$$\|A_\mathcal{S}^\top A_\mathcal{S} - I_{|\mathcal{S}|}\| \leq 0.5. \tag{17}$$

It can be checked easily that the eigenvalues of $A_\mathcal{S}^\top A_\mathcal{S}$ lie between 0.5 and 1.5. Hence, $\kappa\left(A_\mathcal{S}^\top A_\mathcal{S}\right) \leq 3$.

## J. Proof of Theorem 5

*Proof.* By Eq. (16), we know that

$$\mathbb{E}\left\|A_{\widetilde{\mathcal{S}}_t}^\top A_{\widetilde{\mathcal{S}}_t} - I_s\right\|_{\infty\to 1} \leq 40s\left(2c^* + 2\sqrt{2c^*}\right) \leq \frac{s}{8}.$$

By Markov's inequality, we know

$$\mathbb{P}\left\{\left\|A_{\widetilde{\mathcal{S}}_t}^\top A_{\widetilde{\mathcal{S}}_t} - I_s\right\|_{\infty\to 1} \geq \frac{s}{7}\right\} \leq \frac{\mathbb{E}\left\|A_{\widetilde{\mathcal{S}}_t}^\top A_{\widetilde{\mathcal{S}}_t} - I_s\right\|_{\infty\to 1}}{s/7} = \frac{7}{8}.$$

Thus, with probability at least $(1 - \delta)$, we will find a subset of columns so that

$$\left\|A_{\widetilde{\mathcal{S}}_t}^\top A_{\widetilde{\mathcal{S}}_t} - I_s\right\|_{\infty\to 1} \geq \frac{s}{7}, \quad \forall t = 1, \cdots, \lceil\lambda_{\min}(AA^\top)\rceil.$$

From the analysis of Theorem 3 and Theorem 4, we know that for each $t = 1, \cdots, \lceil \lambda_{\min}(AA^\top) \rceil$,

$$|\mathcal{S}_t| = \Theta(k), \quad \text{and} \quad \kappa\left(\sum_{i \in \mathcal{S}_t} \alpha_i^*(\alpha_i^*)^\top\right) \leq 3.$$

Following the same argument as in the proof of Theorem 3, we know that

$$|\mathcal{S}| = \Omega(k\lambda_{\min}(AA^\top)), \quad \text{and} \quad \kappa\left(\sum_{i \in \mathcal{S}} \alpha_i^*(\alpha_i^*)^\top\right) = \Theta(1),$$

satisfying the conditions in Definition 2.

$\square$

## K. Other Supporting Results

### K.1. Proof of stable rank equivalence

*Proof.*

$$\begin{aligned}
\|B^\star A\|_F^2 &= \sum_{i=1}^M \|B^\star \alpha_i^\star\|_2^2 = \sum_{i=1}^M \langle B^\star \alpha_i^\star, B^\star \alpha_i^\star \rangle = \sum_{i=1}^M \text{trace}\left(B^\star \alpha_i^\star (B^\star \alpha_i^\star)^\top\right) \\
&= \sum_{i=1}^M \text{trace}\left(\alpha_i^\star (\alpha_i^\star)^\top (B^\star)^\top B^\star\right) \\
&= \sum_{i=1}^M \text{trace}\left(\alpha_i^\star (\alpha_i^\star)^\top\right) \\
&= \sum_{i=1}^M \|\alpha_i^\star\|_2^2 = M = \|A\|_F^2,
\end{aligned} \tag{18}$$

and

$$\|B^\star A\|^2 = \sup_{x \in \mathcal{S}^{d-1}} x^\top B^\star A A^\top (B^\star)^\top x.$$

Recall that $B^\star \in \mathbb{R}^{d \times k}$ with orthonormal columns. Thus, for each $\widetilde{x} \in \mathcal{S}^{k-1}$, there exists at least one $x \in \mathcal{S}^{d-1}$ such that $(B^\star)^\top x = \widetilde{x}$. Thus,

$$\begin{aligned}
\|B^\star A\|^2 &= \sup_{x \in \mathcal{S}^{d-1}} x^\top B^\star A A^\top (B^\star)^\top x \\
&= \sup_{\widetilde{x} \in \mathcal{S}^{k-1}} \widetilde{x}^\top A A^\top \widetilde{x} \\
&= \|A\|^2.
\end{aligned} \tag{19}$$

Therefore, by Eq. (18), Eq. (19), and the definition of stable rank in Eq. (7),

$$\text{st.rank}(B^\star A) = \text{st.rank}(A).$$

$\square$

