# OpenReview forum: "On the Power of Source Screening for Learning Shared Feature Extractors"
_ICML.cc/2026/Conference — ICML 2026 spotlight_

### Official Review · Reviewer_Vrhb · 2026-02-24

**Soundness:** 2
**Presentation:** 1
**Significance:** 2
**Originality:** 2
**Overall Recommendation:** 2
**Confidence:** 5

**Summary:**

This work studies “source screening” when learning a shared subspace from multiple linear regression models. Statistical properties such as subspace estimation error bound and minimax lower bound are studied. The work proposes a ``gene-aided'' algorithm for the purpose, and proposes approximation approaches to fill the gap left by the ``gene''.

**Compliance With Llm Reviewing Policy:**

Affirmed.

**Final Justification:**

The reviewer had a very difficult time reading and following this manuscript. The notations, the organization, and the terminologies used could use more time and care to revise for accessibility. The setup is restricted to linear models, which, from theoretical understanding viewpoint, is not a fatal aspect. However, this does limits its significance. The discussion in the rebuttal phase is not clear enough to convince the reviewer that the setting can be extended to meaningful nonlinear models, due to character limitations.

**Key Questions For Authors:**

The reviewer does not have specific questions. The overall accessibility should be substantially enhanced for general ML readers.

**Limitations:**

There is not a "limitation" section in the work. Please consider discussing the limitations, e.g., the adoption of relatively unrealistic linear models.

**Strengths And Weaknesses:**

Strengths:

- The work aims to provide algorithms, performance characterizations, as well as validations on simulated and real-world data, presenting a comprehensive research cycle.



Weaknesses:


**Presentation**: The biggest concern lies in presentation clarity. The paper is extremely hard to parse. Notations are not always well defined or even defined at all.  Some examples are as follows:

-- The introduction has a “technical question”, asking if there is S \subset C. But none of S and C are defined in this question or before this question. There’s a notation n in this question, which is also not defined.

-- The terminologies are not easy to follow; they may be used in some areas but may not be well-established conventions in ML, making reading the paper not easy. For example, the so-called “distinct local heads” seem to be the coefficients representing \theta_i in a subspace spanned by B^star. The assumption that such “distinct local heads” are orthonormal is hard to follow and was not explained.

-- The entire section of “the potentials of source screening” is not easy to follow or parse. It is hard to see which source is screened and what is the method to screen it.

-- There is no explanation why Algorithm 1 is “Gene-aided”. It perhaps means that the A matrix is not known in practice. It is also entirely unclear what derivations lead to Algorithm 1. It is also unclear how much the empirically estimated A affect the validity of Algorithm 1.

-- The description of the experiments is also hard to follow. For example, the CelebA experiment is only briefly described. The correspondence to x_ij, y_ij and theta_i in (2) is not clear. It is also unclear to the reviewer what is a “client” in this case.


**Significance**: The considered linear regression model is perhaps far away from reality and is less used in large-scale, complex task-solving machine learning systems these days. It assumes that source i’s covariance has a subspace spanned by B^\star that is also shared by other sources’ covariances. It is also unclear if selecting sources would benefit general applications as the method is validated with small size problems with data dimensions from d=10 to d=192.

**Originality**: The work appears to be largely built upon existing results from Niu et al 2024. Due to the clarity challenges, it is hard to assess how much new innovation is presented in this current work beyond Niu et al 2024.

---

> ### Author Rebuttal · Authors · 2026-03-30
>
> We thank the reviewer for the constructive feedback. We have added an overview to Section 4 to clarify each subsection's role and ensured all notation is precisely defined.
>
> **W1:** Thanks for pointing this out. We now establish this notation prior to the technical question.
>
> **W2:** We agree the notation for distinct local heads ($\hat{\alpha}\_1^{* }, \cdots , \hat{\alpha}\_k^{* }$) may cause confusion with the local heads $\alpha_j^{* }$. In the revision, we replace them with $\nu_1, \cdots, \nu_k$. Please note that the restricted, orthogonal local head values are assumed only  in Section 4.1. This special family simplifies computing the eigenvalues and eigenvectors of matrix $D$, serving purely as a motivating example to build intuition on how source screening reduces error rates. The remaining subsections in Section 4 consider a general collection of $\alpha_j^{* }$.
>
> **W3:** We thank the reviewer for pointing this out. We have revised and will further revise Section 4.1 to add explanations.
>  The purpose of Section~4.1 is to provide intuition for why and how much source screening can help in improving the statistical error rates. In that motivating example, sources belong to $k$ different groups, and these groups may be highly imbalanced in size. If one trains on the full population, then the aggregate matrix formed by the local heads can be poorly conditioned because some directions are overrepresented while others are underrepresented, which makes the shared representation harder to estimate accurately. In this example, source screening can be performed by selecting an arbitrary subset of  $m_k/n$ sources for each of the $k$ directions, recalling that $m_k/n$ denotes the number of sources with $\alpha_i^* = \mu_k$​. We have emphasized this point in the revision.
>
> **W4:** We clarified "genie-aided" as denoting Algorithm 1's access to the unobservable matrix $A=[\alpha_1^{* },\dots,\alpha_M^{* }]$ to establish theoretical limits, whereas Algorithm 2 uses practical data-driven estimates. Algorithm 1 is the direct implementation of Theorems 3 and 4, which collectively prove that selecting a well-conditioned subset of columns translates to an admissible subpopulation with improved shared-subspace estimation.
> Our experiments in Section 5.1 compare our empirical algorithm with the genie-aided version, with both methods outperforming full population training.
>
> **W5:**
> We agree that the experimental setup should be described more concretely. To ensure consistency, we now exclusively use the term "source" rather than "client" and have detailed the setup in Section 5.2. In both experiments, a source is a natural data partition. In ACSIncome, a source is a US state with raw tabular features $x_{ij} \in \mathbb{R}^{10}$. In CelebA, a source is a distinct celebrity, where $x_{ij} \in \mathbb{R}^{192}$ represents the embedding of an image of that celebrity extracted using a pre-trained Vision Transformer. For both, $y_{ij}$ is the label of the $j$-th sample for source $i$, and $\theta^*_i$ is the source-specific parameter.
>
> **Significance**
> Thank you very much for sharing your concern. While our current theoretical results are developed for linear models, the framework is likely to extend beyond this setting. We refer to our response to W1 of Reviewer 7TFn for further discussion. In this work, our intent is to use the linear model as a standard theoretical abstraction that permits precise analysis, while testing the source-screening principle empirically in more realistic settings.
>
> In our applied experiments, our screening method is used as a pre-processing step to identify high-quality sources prior to initiating the FedRep training regime, which is not limited to linear models.
> To further validate this, we conducted additional experiments on the ACSIncome dataset (n=1.6m) where, in place of linear regression, we train a two-layer neural network as a nonlinear feature extractor. As shown in the table below, our screening procedure consistently benefits the training of these nonlinear models across different hidden layer sizes (d=64 and d=256).
>
> | Selection Method | Mean Accuracy ($d=64$) | Mean Accuracy ($d=256$) |
> | :--- | :---: | :---: |
> | **Empirical (Ours)** | **80.13%** | **79.96%** |
> | All Clients | 79.91% | 79.49% |
> | Power-of-Choice | 79.70% | 79.41% |
> | Random Selection | 78.08% | 77.87% |
>
> **Originality**
> Thank you for sharing this concern. We have clarified the distinction between our work and Niu et al. (2024). We cite them because their upper and lower statistical bounds represent the state-of-the-art. However, our contributions are fundamentally different: while they analyze subspace training, we identify and characterize when selective source screening improves shared-representation learning. Although we apply their split-averaging estimator to our selected subpopulation, our results are agnostic and valid for any minimax-optimal estimator applied to a well-conditioned $D$.

---

> > ### Author Rebuttal · Reviewer_Vrhb · 2026-04-02
> >
> > I thank the authors for the their reply and clarifications. However, I cannot support publishing this work due to the clarity issues. The work could use more time to enhance its accessibility. Regarding significance beyond linear models, "likely be able to extend beyond this setting" is not sufficiently convincing.

---

> > > ### Author Response · Authors · 2026-04-03
> > >
> > > **Accessibility:**  We thank the reviewer for acknowledging our clarifications. We agree that improving accessibility is important. As detailed in our rebuttal, we have identified concrete revisions that will be incorporated into the camera-ready version, and we will conduct multiple rounds of revision to further enhance readability. If helpful and compatible with the ICML rules, we would also be happy to share a revised version reflecting these changes for the reviewer’s reference.
> > >
> > > **Beyond linear models:**
> > > Thank you for raising concerns about the extension to nonlinear models. We agree that the phrase “likely be able to extend” is imprecise and will revise it. Due to space constraints, some details were omitted in our original response; we provide further clarification below.
> > >
> > > In our submission, Theorems 3, 4, and 5 do not depend on the linearity of the data-generating models. In particular, $\alpha_i^*$ need not take the linear form in Eq. (3). These results hold as long as the matrix $A$ satisfies the regularity conditions specified in the corresponding theorems.
> > >
> > > It remains to verify Theorem 2, which concerns minimax optimality rates. Both the upper and lower bounds depend on the data-generating model, as well as distributional and other technical assumptions. For the general nonlinear setup in Section 7 of [Niu et al., 2024] (also referenced in our response to comment W1 by reviewer 7TFn), assume: (1) $x_{ij} \sim \mathcal{N}(0, I_d)$ i.i.d. for all $j = 1, \ldots, n$ and $i = 1, \ldots, M$, and (2) Assumption 7.2 in [Niu et al., 2024]. Under these conditions, one can obtain an upper bound with the same dependence on the spectrum of the matrix $D$ defined in Eq. (4), with the key difference that $\alpha_i^* $ takes the form $\alpha_i^* = \mathbb{E}_{U \sim \mathcal{N}(0, I_k)}[h_i(U)U]$, as shown in [Niu et al., 2024] immediately following Lemma 7.1. Here, $h_i$ is the general source-specific head in our Eq.(1). Moreover, the same lower bound continues to hold under assumptions (1) and (2). By applying the same argument as in Theorem 2, the result extends to this nonlinear model.
> > >
> > > We will add this extension in a revision.

---

### Official Review · Reviewer_NSGP · 2026-03-02

**Soundness:** 3
**Presentation:** 3
**Significance:** 3
**Originality:** 3
**Overall Recommendation:** 5
**Confidence:** 5

**Summary:**

The paper challenges the conventional wisdom in multi-task and representation learning, which suggests that incorporating more relevant data sources always leads to a better shared feature extractor. The authors investigate the problem of Source Screening within a collection of good sources.
Through a rigorous theoretical framework, the paper demonstrates that even when all sources are relevant to the underlying common structure, learning from a carefully selected subset can be more powerful than using the entire collection. The authors provide theoretical guarantees regarding the sample complexity and the quality of the learned representation when screening is applied, particularly focusing on the spectral properties of the source distributions.

**Compliance With Llm Reviewing Policy:**

Affirmed.

**Key Questions For Authors:**

In a real-world scenario where the ground truth parameters are unknown, what proxy metrics do you suggest for selecting the subset? Can this be done in a single pass or through an incremental learning approach?
Does the Power of Source Screening still hold when the shared feature extractor is a high-capacity non-linear network? Do you expect the redundancy of sources to increase or decrease in such spaces?
How sensitive is the framework to the condition? If the sources are highly aligned, does the screening gain vanish?

**Limitations:**

The study focuses on good sources to isolate the screening effect. However, in practice, the presence of bad sources is a major confounder. The interaction between filtering noise and screening relevance is not fully explored.
The current framework assumes a fixed set of sources. In modern Cloud Computing environments, sources often arrive in a stream. The paper does not address how the optimal screened subset should evolve over time.
The power is defined in terms of representation error, but the paper lacks a discussion on the trade-off between screening time and training time, which is critical for practitioners.

**Strengths And Weaknesses:**

The core thesis—that a subset of relevant sources can outperform the full set—is a significant and non-obvious contribution to learning theory. It provides a formal justification for data pruning in representation learning. The use of spectral analysis to characterize the power of a source collection is technically sound and provides a clear metric for source diversity. Unlike many papers that assume sources, this work explicitly models the heterogeneity across sources, making it highly relevant to real-world big data scenarios where data originates from diverse silos. The paper makes an important distinction between filtering out noisy data and screening redundant but otherwise good data.
While the theoretical analysis of Theorem 3 and 4 is compelling, the paper is somewhat light on a practical, scalable algorithm for identifying the optimal subset in high-dimensional settings without prior knowledge of the true feature extractor. The theoretical derivations appear to rely heavily on linear representation learning. It is unclear how well these guarantees generalize to non-linear feature extractors like Deep Neural Networks. The paper focuses on the power of the resulting extractor but does not fully account for the computational search cost of screening itself. If the screening process is as expensive as training on the full set, its practical utility in Big Data environments may be diminished.

---

> ### Author Rebuttal · Authors · 2026-03-31
>
> **W1:** (The theoretical derivations appear to rely heavily on linear representation learning. It is unclear how well these guarantees generalize to non-linear feature extractors like Deep Neural Networks.)
>
> Thanks for your comments. While our current theoretical results are developed for linear models, the framework is likely to extend beyond this setting. We refer to our response to W1 of Reviewer 7TFn for further discussion. For convenience, we briefly restate the key points below.
> - The analysis could potentially be generalized to a broader class of nonlinear models of a general form that incorporates neural networks (one-hidden-layer architectures) as a special case.
> - Under stronger regularity assumptions on the covariate distributions, the vector $\mathbb{E}\_{U\sim N(0, I_k)}[h_i(U)U]$ acts as $\alpha_j^{* }$ in the linear model. The matrix D that captures collective heterogeneity takes the form $D = \frac{1}{M}\sum_{i=1}^M \mathbb{E}\_{U\sim N(0, I_k)}[h_i(U)U]\mathbb{E}\_{U\sim N(0, I_k)}[h_i(U)U^{\top}]$.
> - Since our theories are mainly derived from improving the condition numbers of $D$ and $A$ rather than the particular linear form of $h_i$, we expect our approach to extend to this broader setting.
>
> **W2:** (Clarification on the computational search cost.)
>
> We thank the reviewer for this comment. We emphasize that the practical benefit of our method is twofold: beyond computational savings from training on fewer sources, screening can also yield strictly better model quality in heterogeneous settings, as demonstrated in our experiments. Moreover, the screening step itself is lightweight, taking less time than a round of training. For example, on ACSIncome, our screening procedure takes approximately 2 seconds, compared to roughly 7 minutes to complete full population training. We will include an extended runtime comparison table in the revised version.
>
> | Dataset | Data Screening | Avg. Training Time/Round (Ours) | Avg. Training Time/Round (Full Populationl) |
> | :--- | :---: | :---: | :---: |
> | **ACSIncome** | 1.60 s | 4.79 s | 34.94 s |
> | **CelebA** | 12.78 s | 22.35 s | 148.15 s |
>
> **Q1:** (Subset selection and proxy metrics with unknown ground truth parameters)
>
> Indeed, in practice we do not have access to ground truth $\alpha_i^{* }$ to perform screening. In section 4.3 we present practical heuristics to circumvent the need for this information, and our experiments show competitive performance of our empirical selection algorithm compared to the genie-aided case.
>
> **Q2:** (On the benefits of source screening with neural network feature extractor)
>
> We thank the reviewer for their question. To further validate this, we conducted additional experiments on the ACSIncome dataset where, in place of linear regression, we train a two-layer neural network as a nonlinear feature extractor. As shown in the table below, our screening procedure consistently benefits the training of these nonlinear models across different hidden layer sizes (d=64 and d=256).
>
> | Selection Method | Mean Accuracy ($d=64$) | Mean Accuracy ($d=256$) |
> | :--- | :---: | :---: |
> | **Empirical (Ours)** | **80.13%** | **79.96%** |
> | All Clients | 79.91% | 79.49% |
> | Power-of-Choice | 79.70% | 79.41% |
> | Random Selection | 78.08% | 77.87% |

---

### Official Review · Reviewer_7TFn · 2026-03-11

**Soundness:** 4
**Presentation:** 3
**Significance:** 2
**Originality:** 4
**Overall Recommendation:** 5
**Confidence:** 3

**Summary:**

The paper presents theoretical results showing that instead of using all the available data in a linear feature extraction problem, a suitably selected subset could potentially result in a better and statistically optimal minimax error. Particularly, it is shown that we should have balanced data from different classes to achieve optimal results. It is essentially demonstrated that by discarding data from classes with more data, the overall performance improves (which is counter-intuitive). Besides, a randomized method is proposed to estimate this optimal subset.

**Compliance With Llm Reviewing Policy:**

Affirmed.

**Final Justification:**

The rebuttal reinforced my prior positive assessment of the paper’s theoretical soundness and presentation. The authors addressed the theoretical and structural questions well. However, the fact remains that the empirical improvements on real-world datasets are marginal.

According to ICML reviewing guidelines, raising the score to a 6 would require the paper to be essentially flawless. Given the acknowledged empirical limitations, a higher score is not warranted. To strike a proper balance, I maintain my initial score. The paper is theoretically solid, introduces a sound methodology, and offers enough significance to warrant publication. Therefore, I believe a 5 (Accept) remains the most appropriate and fair recommendation. I strongly encourage the authors to incorporate their rebuttal discussions (especially regarding nonlinear generalization and notation corrections) into the final camera-ready version.

**Key Questions For Authors:**

1. Regarding the notations: the norm in Definition 1 is not mentioned.
2. The $\kappa$ notation for the condition number is not introduced.
3. Section 4 starts with the special example where the local heads are mutually orthogonal, which is very restrictive. However, after that (e.g., in Subsection 4.2), it is not stated whether this restriction is still valid or not.
4. I think the proof of Theorem 3 through the Kadison-Singer problem is simpler.
5. In the general setting, the $\hat{\alpha}^*$'s can all be different, even though they lie in the $k$-dimensional space. What happens in this case?

**Limitations:**

1. I guess I mentioned them all in my previous comments.

**Strengths And Weaknesses:**

**Strengths:**
1. The general message is very interesting, and the theoretical arguments seem valid.
2. A practical method is proposed which could be helpful in real applications.

**Weaknesses:**
1. The theories are restricted to linear models.
2. The notations are, in some cases, not properly defined.
3. The improvement in the case of real data is marginal.

---

> ### Author Rebuttal · Authors · 2026-03-31
>
> **W1:**
> Thanks for your comments. While our current theoretical results are developed for linear models, the framework is likely to be generalized to a broader class of nonlinear models of the general form $ \mathbb{E}[y_{ij} \mid x_{ij}]= h_i((B^{* })^{\top}x_{ij})$, where $h_i$ is a source-specific function. That is, one can specialize $\phi$ in Eq.(1) of our submission to be $B^*$.  This family incorporates canonical nonlinear models such as generalized linear models and one-hidden-layer neural networks, as shown in Section 7 of [Niu et al., 2024].
>
> Under stronger regularity assumptions on the covariate distributions (particularly standard Gaussian on $x_{ij}$ for all $i$ and all $j$), the vector $\mathbb{E}\_{U\sim N(0, I_k)}[h_i(U)U]$ in the linear model. The matrix D that captures collective heterogeneity takes the form $D = \frac{1}{M}\sum_{i=1}^M \mathbb{E}\_{U\sim N(0, I_k)}[h_i(U)U]\mathbb{E}\_{U\sim N(0, I_k)}[h_i(U)U^{\top}]$.
>
> Since the linear model is a special case of this general family, existing minimax lower bounds on linear models still hold. It remains to verify whether, for a given subpopulation of sources, similar upper bounds in terms of the eigenvalues of $D$ still hold. As formally argued in [Niu et al., 2024], such upper bounds can be established using similar split-averaging techniques. Since our theories are mainly derived from improving the condition numbers of $D$ and $A$ rather than the particular linear form of $h_i$. Hence, we expect our approach to extend to this broader setting.
>
> **W2:**
> We agree that the notation can be improved. In our revision, we will address the specific issues raised (detailed in Q1 and Q2) and perform a comprehensive pass to enhance readability across the main text and proofs.
>
> W3:
> We agree there is room to improve real-data performance. In this paper, we aim to develop a new theoretical perspective on existing feature extractor training; the real-data experiments are mainly a proof of concept. While the current numerical gains are modest, we believe they are consistent and indicative of the potential of the approach.  As these results rely on heuristic choices in both the algorithms and the experimental setup, there remains significant room to improve empirical performance in future work.
>
>
> **Q1/Q2:** We thank the reviewer for pointing this out. We now explicitly specify the spectral norm in Definition 1, and introduce $\kappa(\cdot)$ as the condition number upon its first appearance.
>
> **Q3:**
> We thank the reviewer for pointing this out. The mutually orthogonal local-head assumption is only used in Section 4.1 as a motivating special case to build intuition on the potential of source screening. It is not imposed in Section 4.2. In particular, Section 4.2 explicitly studies the question for a general collection of local-head configurations $\{\alpha_i^\ast\}_{i=1}^M$, and the assumption that $A=[\alpha_1^\ast,\dots,\alpha_M^\ast]$ is standardized means only that each column has unit norm, not that the columns are orthogonal. We agree that this transition should be stated more clearly, and we will revise the manuscript to make this explicit at the beginning of Section 4.2.
>
> **Q4:**
> We thank the reviewer for this insightful suggestion and for drawing our attention to the Kadison–Singer problem. At present, we do not have sufficient background to say definitively if it may be applied to our setting, but we hope to explore this connection in future work.
>
> Nevertheless, during the preparation of this response, we came across Weaver’s conjecture and the Marcus–Spielman–Srivastava theorem [1], which resolves Kadison–Singer for finite-dimensional matrices. Due to space limitations, we will provide the technical connections in the follow-up response if that helps.
>
> [1] Marcus, Spielman, and Srivastava, Interlacing Families II: Mixed Characteristic Polynomials and the Kadison–Singer Problem, Annals of Mathematics 182 (2015)
>
>
> **Q5:**
> Thank you for raising this point and clarify that the $k$ distinct, mutually orthogonal local heads in Section 4.1 serve purely as a motivating example to build intuition and enable a closed-form eigenvalue analysis of $D$. It is not the setting our algorithms are designed for or evaluated on. Our framework handles the fully general case where distinct $\alpha_i^{* }$ lie anywhere in the $k$-dimensional subspace, with learnability governed by the diversity matrix $D = \frac{1}{M} \sum_i \alpha_i^{* } (\alpha_i^{* })^\top$ to accommodate arbitrary heterogeneity. We evaluate this directly in Section 5.1's Heterogeneous Gaussian setting, where drawing each $\alpha_i$ from a client-specific covariance $\Psi_i$ yields the distinct, unstructured $\alpha_i^{* }$ that our algorithm successfully handles, and we will add a clarifying remark to make this scope explicit.

---

> > ### Author Rebuttal · Reviewer_7TFn · 2026-04-03
> >
> > Thank you to the authors for their detailed rebuttal. Most of my concerns have been adequately addressed. However, the improvement on real-world datasets remains marginal, which is still not fully resolved in my view.
> >
> > Overall, I continue to find the paper interesting. The theoretical arguments appear valid, and the proposed practical method could be helpful in real applications.
> >
> > To strike a proper balance between the original submission and the rebuttal (in which the authors addressed most of my questions), I recommend acceptance with the following suggestions for the final version (if the paper is accepted):
> >
> > - Add a discussion of the potential generalization to a broader class of nonlinear models (as briefly outlined in the rebuttal) as future work.
> > - Carefully correct the remaining notation and revision issues throughout the paper.
> >
> > Because the rebuttal was satisfactory, I have not decreased my score. At the same time, given that the original paper still has the limitations mentioned above, I am not able to raise the score to 6. According to ICML guidelines, a score of 6 would require the paper to be essentially flawless. Therefore, I believe 5 (Accept) remains the most appropriate score for this submission.

---

### Official Review · Reviewer_X7nh · 2026-03-11

**Soundness:** 4
**Presentation:** 4
**Significance:** 3
**Originality:** 3
**Overall Recommendation:** 5
**Confidence:** 3

**Summary:**

The paper considers the problem of shared representation learning in the linear setting. The authors argue that having clients which are equally spread along different local heads are more suited for learning and give better error bounds than having clients with highly skewed local heads. The authors motivate this through a toy setting and then formalize the existence of a sub-population with equally distributed heads. Finally, the authors present their algorithm in the genie-aided setting (assuming access to local heads) and then relax this condition through heuristics.

**Compliance With Llm Reviewing Policy:**

Affirmed.

**Key Questions For Authors:**

1. Does the analysis still hold true if the clients are appropriately re-weighted instead of being removed?
2. How would the task of MTL be instantiated for (say) downstream LLM finetuning?

**Limitations:**

yes

**Strengths And Weaknesses:**

**Strenghts**
1. The problem is interesting and the method is sound
2. The paper is well presented.
3. The paper presents some interesting insights, formalizing source screening in the scenario where all sources are clean.

---

> ### Author Rebuttal · Authors · 2026-03-30
>
> **Q1:**
> Thank you very much for your question. It would be interesting to explore this direction further. A preliminary examination suggests that the analysis does not immediately carry over. Below, we outline some initial thoughts on the analytical challenges. Due to time constraints, these ideas are not fully developed. Please feel free to let us know if we have overlooked anything.
> 1. Existing upper bounds do not readily extend to sample reweighting.  To see this, consider the special class of problem instances in Section 4.1. Appropriate reweighting $\alpha_j^{* }$ will easily make the reweighted diversity matrix D well-conditioned. However, reweighting also directly impacts the behavior of subspace estimators. In particular, existing analysis of those estimators relies on the statistical concentration of all used samples as a whole, a structure that sample reweighting may undermine.
> 2. For the general collection of $\alpha_j^{* }$, departing from the results in Section 4.2, appropriate sample reweighting may not exist. Existing analysis in Section 4.2 heavily relies on (i) column selection (Algorithm 1: line 12 - line 20 ) and (ii) the concatenation of the selected columns (Algorithm 1: line 29). Sample reweighting will significantly affect the correctness of both (i) and (ii). Roughly speaking, assuming matrix A is standardized (i.e., all columns have l2 norm equal to 1),  Theorem 4 guarantees that, at each iteration, the outer product of the selected columns, taken as a whole, has a constant-level condition number. Sample reweighting undermines this column standardization structure, affecting the construction of the matrix H in line 943 (Appendix G.1). In particular,  instead of subtracting $A^{\top}A$ by $I_M$, one may need to subtract a diagonal matrix with entries that are closely related to the weights. Following the roadmap of existing analysis, the condition number of the weighted version of $A_S^{\top} A_S$ may no longer remain constant. Moreover, unless a more refined version of Weyl’s inequalities is available, the accumulation of weights (analogous to matrix concatenation in the original analysis) may fail to preserve a constant condition number.
>
> Numerically, sample reweighting may help to reduce noise compared with pure removal. We train a two-layer neural network on the ACS Income Dataset and evaluate this reweighting scheme as an alternative to client subsampling. Specifically, we compute the Grothendieck Factorization of the matrix $\hat{A}^{T}\hat{A} - I_M$ (i.e., line 19 of our empirical subpopulation search, but applied to all clients), and we use the inverse of the client $d$ values to weight their contribution during distributed averaging with FedRep. We consider two paradigms: V1, which only considers $d$ with weights $\frac{d_j^{-1}}{\sum_{j’} d_{j’}^{-1}}$, and V2, which also incorporates client data volume $m_j$ (as in standard federated averaging) with weights $\frac{m_j d_j^{-1}}{\sum_{j’} m_{j’} d_{j’}^{-1}}$.
> We find that while the reweighting strategies still outperform full population training with lower variance across seeds, they have a lower average accuracy than source subsampling.
>
> | Selection Method | Avg | Std |
> | :--- | :--- | :--- |
> | Subsampling | 79.96 | 0.21 |
> | Reweight V1 | 79.38 | 0.03 |
> | Reweight V2 | 79.58 | 0.06 |
> | Full Population | 79.49 | 0.06 |
>
> **Q2:**
> Our formulation can be naturally interpreted in the context of LLM finetuning. Taking the pre-trained LLM as fixed, one can view our setup as learning a shared adapter layer (a generic dense matrix appended to the pre-trained LLM backbone) alongside task-specific classification heads. The shared adapter corresponds to our shared subspace $B^{* }$, and the task-specific heads correspond to the $\alpha_j^{* }$. Our source screening procedure would then select which finetuning tasks to include when learning this adapter. We also note that our CelebA experiments already reflect a version of this paradigm, where a pre-trained vision transformer is held fixed, and only the downstream heads are learned. In this case, $B^*$ is a shared $d \times{} k$ matrix which reduces the dimension of the pre-trained feature from $d$ to $k$ prior to the client headers. Extending the theoretical analysis to other finetuning strategies, such as prompt tuning or LoRA, is an exciting direction for future work.

---

> > ### Author Rebuttal · Reviewer_X7nh · 2026-04-02
> >
> > I would like to thank the authors for their rebuttal. I am keeping my score.

---

### Decision · Program_Chairs · 2026-04-30

**Decision:**

Accept (spotlight)

**Comment:**

This paper shows that in the meta-learning setup, using a subset of sources for representation learning can lead to better estimators than by using the entire data that is available. Indeed, the reviewers agree that the paper has a strong message with good underlying theoretical guarantees. In particular, I myself liked the toy example quite a bit.

One reviewer was concerned about clarity issues - I read the paper myself and indeed, it is quite evident. Several notations remained undefined - for instance, In Section 3, N is the total number of datapoints but it has never been defined properly. Unfortunately the paper is scattered with such notational issues. Given that I am well acquainted with this line of work, I was able to follow the setup clearly - however, I can also understand why the writing is very terse and complex to follow (notations do not help) for someone who is reading the paper for the first time.

I do not think that any of the clarity issues is deep - all of them can be fixed in the camera-ready. I urge the authors to provide particular attention to notations, clarity and explanation in the final version.